# U-shaped and Inverted-U Scaling behind Emergent Abilities of Large Language Models

**Tung-Yu Wu, Melody Lo**
National Taiwan University
{b08901133, peiyulo}@ntu.edu.tw

## Abstract

Large language models (LLMs) have been shown to exhibit *emergent abilities* in some downstream tasks, where model performance stagnates at first and then improves sharply and unpredictably with scale beyond a threshold. In this work, we investigate the phenomenon by grouping questions based on difficulty level and provide a possible explanation for emergent abilities. Specifically, we observe U-shaped scaling for hard questions and inverted-U scaling followed by steady improvement for easy questions. The two scaling patterns initially offset each other, causing stagnant overall performance. The performance starts to soar when the scaling pattern of easy questions reverts from inverse to standard scaling, leading to emergent abilities. Based on this finding, we propose a simple yet effective pipeline, called *Slice-and-Sandwich*, to predict the emergence threshold and model performance beyond the threshold. Our code is publicly available at https://github.com/tony10101105/ExpEmergence.

## 1 Introduction

Large language models (LLMs) (Team et al., 2023; Achiam et al., 2023; Brown, 2020; Touvron et al., 2023a;b; Workshop et al., 2022; Li et al., 2023; Jiang et al., 2024) have shown strong potential in various downstream applications (Jumper et al., 2021; Fawzi et al., 2022; Naveed et al., 2023; Kaddour et al., 2023). Though the training-loss scaling law has been well established (Kaplan et al., 2020; Hoffmann et al., 2022), the literature is inconclusive regarding how performance on downstream tasks scales. In particular, for certain downstream tasks (Srivastava et al., 2023; Lin et al., 2022a; Pilehvar & Camacho-Collados, 2019), LLMs display *emergent abilities*: performance stagnates even when model training compute scales up hundredfold, and then improves sharply at an unpredictable critical threshold (Wei et al., 2022; Schaeffer et al., 2024a).

Some prior work (Schaeffer et al., 2024a;b; Lu et al., 2024) link emergent abilities to crude performance metrics that fail to capture small model improvements. Hu et al. (2023) introduces the *PASSUNTIL* metric, showing gradual model improvement with scale. Schaeffer et al. (2024a) finds that emergent abilities mainly happen on string-match and multiple-choice tasks (Schaeffer et al., 2024a), for which traditional performance measures exhibit strong discontinuity. They propose continuous metrics such as Brier Score (Brier, 1950) and linear metrics such as token edit distance (TED) (Schaeffer et al., 2024a) to better predict LLM scaling law on downstream tasks. Schaeffer et al. (2024b) further ranks several performance metrics in correlation with the model scale. On the other hand, Michaud et al. (2024) establish the quantization model of neural scaling to explain the emergent drop of cross-entropy loss from the aspect of next-token prediction.

Another focus of prior literature is the predictability of emergent abilities measured in traditional metrics like accuracy, which is crucial for monitoring and forecasting LLMs' potentially harmful task-wise capabilities, such as writing certain malicious code (Charan et al., 2023). Though Wei et al. (2022) characterizes emergent abilities as *unpredictable* performance soar, some studies (Ruan et al., 2024; Gadre et al., 2024; Hu et al., 2023; Owen, 2024; Ye et al., 2023) have proposed pipelines to estimate task-specific scaling law. However, they usually incorporate models past the emergence threshold into the training set to fit a Sigmoid function and do not predict emergence abilities.

This paper contributes to both fronts of discussion on emergent abilities, especially for multiple-choice tasks. First, we propose a novel procedure to evaluate the performance of LLMs on questions grouped

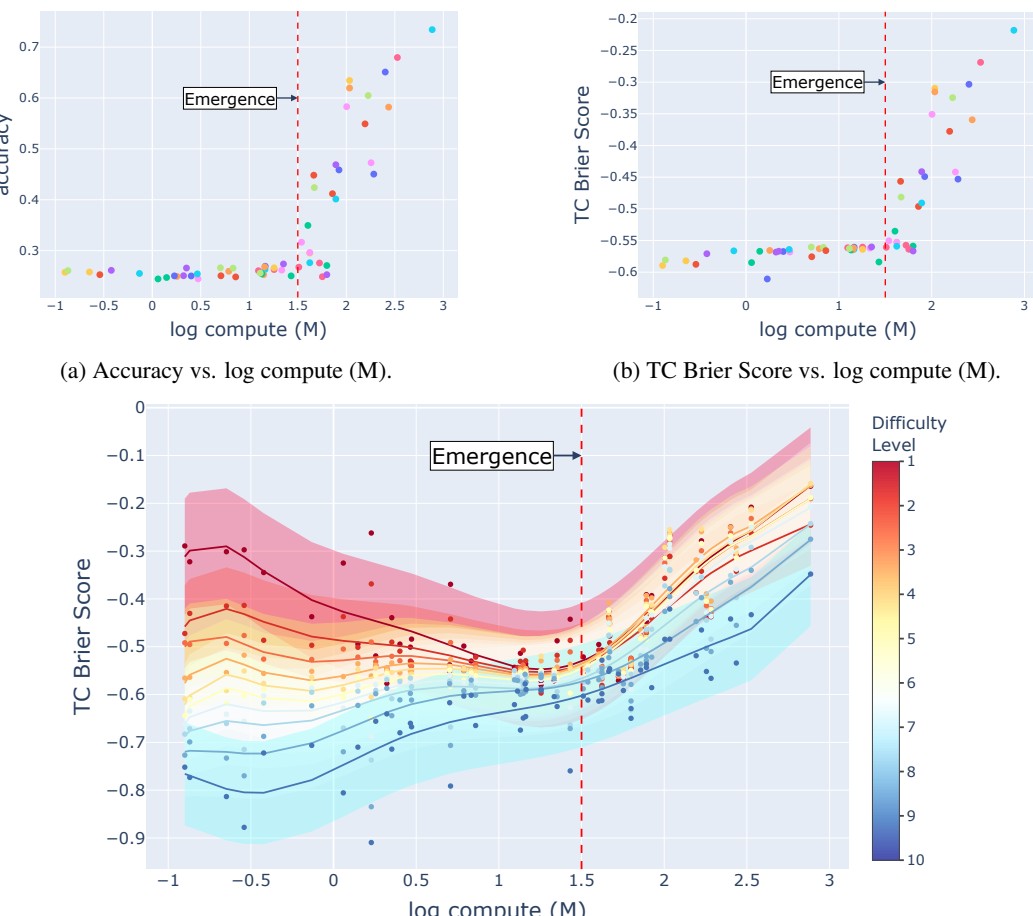

(a) Accuracy vs. log compute (M).     (b) TC Brier Score vs. log compute (M).

(c) U-Shaped and inverted-U scaling with MMLU's questions clustered into 10 groups. Higher levels are harder questions.

Figure 1: The accuracy, Target-Conditioned (TC) Brier Score, U-shaped and inverted-U scaling on the MMLU benchmark (Hendrycks et al., 2021) evaluated using 56 LLMs. Sec. 2.1.2 provides details on the TC Brier Score, which captures granular changes in model performance. App. A provides details of the 56 LLMs.

by different difficulty levels. Fig. 1 shows the evaluation result of 56 LLMs with diverse training compute on the MMLU benchmark, whose 14042 questions are clustered into 10 groups based on their difficulty levels, with higher levels denoting harder questions. Both model performance and difficulty of each question are measured and calculated by the *Target-Conditioned (TC) Brier Score* (see Sec. 2.1.2 for details), which is our proposed continuous metric that is highly correlated with accuracy but can capture more granular changes in model predictions. We observe that performance on hard questions exhibits U-shaped scaling (Wei et al., 2023; McKenzie et al., 2023), where it worsens with scale at first and then reverses to improve with scale. In contrast, performance on easy questions exhibits an inverted U-shape followed by steady improvement with scale, consistent with the previously reported deep double descent of testing loss (Nakkiran et al., 2021). Moreover, the point at which performance reverts from inverse to standard scaling roughly coincides with the emergence threshold beyond which model performance begins to soar. Our observation could explain why LLM's performance on some multiple-choice tasks stagnates for models below the emergence threshold: the scaling trend on easy questions offsets that on hard questions.

This observation of U-shaped and inverted-U scaling provides a basis to predict the forthcoming sharp increase in model performance, a defining feature of emergent abilities. We propose *Slice-and-Sandwich* pipeline, where we first group questions on a given downstream task by difficulty levels, use data before the emergence threshold to fit the performance on easy and hard questions separately,

then forecast performance on easy and hard questions separately beyond the emergence threshold. We show that *Slice-and-Sandwich* captures the performance soar well.

We summarize our contributions as follows:

- We demonstrate that, for some downstream tasks previously shown to display emergent abilities, under a proper continuous metric, LLM's performance exhibits opposing scaling trends: inverted-U vs. U-shape, on easy vs. hard questions below the emergence threshold, and steadily improves beyond the emergence threshold.
- Based on the observation of inverted-U vs. U-shape on easy vs. hard questions, we propose a simple yet effective pipeline, *Slice-and-Sandwich*, to forecast model performance past the emergence threshold. Experimental results on three iconic datasets show its effectiveness.

## 2 SCALING TREND BY DIFFICULTY LEVEL: U-SHAPE VS. INVERTED-U

This section documents LLM's scaling trend by question difficulty level. Sec. 2.1 defines terminologies such as log compute, emergence threshold, and our performance metrics. Sec. 2.2 describes how we group questions by difficulty level. Sec. 2.3 presents and discusses the results of 6 iconic multiple-choice tasks with emergent abilities.

### 2.1 TERMINOLOGY

#### 2.1.1 LOG COMPUTE AND EMERGENCE THRESHOLD

For clearer visualization, in this paper, we refer to an LLM's log compute $M$ as:

$$M = \log_{10}\left(\frac{C}{10^{21}}\right), \tag{1}$$

where $C \approx 6ND$ (Kaplan et al., 2020) is the total training compute (FLOPs) of an LLM, $N$ is the number of model parameters, and $D$ is the number of training tokens. The emergence threshold $T$ is identified manually as the log compute where the model accuracy exhibits a sharp improvement, as illustrated in Fig. 1a.

#### 2.1.2 CONTINUOUS PERFORMANCE METRICS

Prior work (Schaeffer et al., 2024a;b; Lu et al., 2024) has advocated for performance metrics that distinguish finer differences. One candidate metric is the Brier Score (Brier, 1950):

$$Brier = \frac{1}{K}\sum_{t=1}^{K}\sum_{i=1}^{C}\left(\hat{p}_{t,i} - p_{t,i}\right)^2, \tag{2}$$

where $K$ is the number of samples and $C$ is the number of classes. $p_{t,i}$ is 1 if the $t$-th sample belongs to class $i$, and 0 otherwise. $\hat{p}_{t,i}$ is the model's predicted probability of the $t$-th sample being of class $i$.

However, the Brier Score depends not only on the model's predicted probability of the target class (choice) but also on the predicted probability distribution of all classes. Since there is no a priori reason which type of distribution on non-target classes signify better ability [1], we propose the *Target-Conditioned (TC) Brier Score* that lumps all non-target available classes into one class, conditioned on available classes, and has an opposite sign to Eq. 2 so that higher score means higher performance:

$$TC\_Brier = -\frac{1}{K}\sum_{t=1}^{K}\left(\hat{p}_{t,c}^{con} - 1\right)^2, \tag{3}$$

where $\hat{p}_{t,c}^{con}$ is the model's predicted probability of the $t$-th sample being of target class $c$ conditional on available classes, i.e.,

$$\hat{p}_{t,c}^{con} = \frac{\hat{p}_{t,c}}{\displaystyle\sum_{i\in\text{available classes}}\hat{p}_{t,i}}, \tag{4}$$

---

[1]Schaeffer et al. (2024b) found that $\hat{p}_{t,c}^{con}$ (denoted as $p_\theta^{Choices}$(correct choice) in Schaeffer et al. (2024b))) has higher correlation with log compute than Brier Score does.

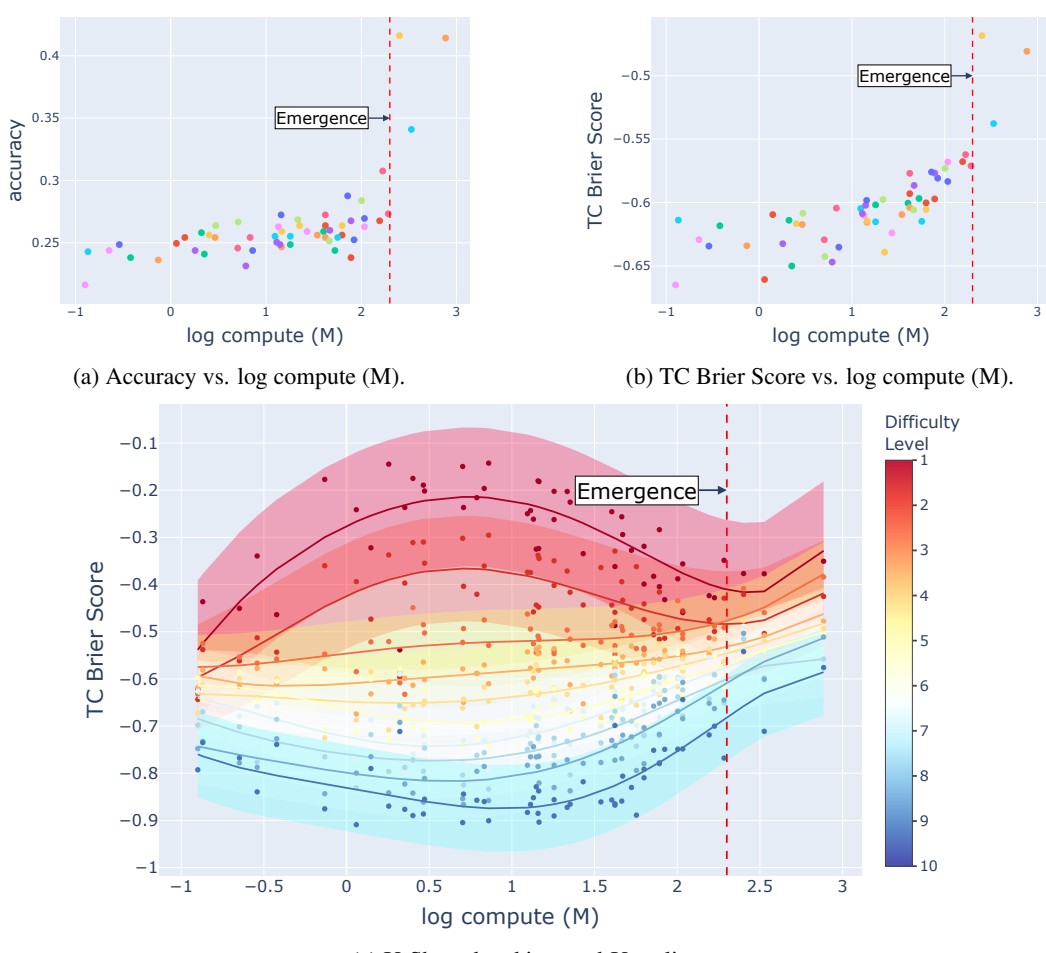

(a) Accuracy vs. log compute (M).

(b) TC Brier Score vs. log compute (M).

(c) U-Shaped and inverted-U scaling.

Figure 2: The accuracy, TC Brier Score, U-Shaped and inverted-U scaling on the Persian-QA dataset in BIG-bench (Srivastava et al., 2023).

where c is the target class, $\hat{p}_{t,c}$ is the model's predicted probability of the $t$-th sample being of the target class. We discuss the effect of conditioning in App. B.

## 2.2 GROUPING QUESTIONS BY DIFFICULTY LEVELS

### 2.2.1 MEASURING QUESTION DIFFICULTY LEVEL

For a question $q$ of a downstream task with emergence threshold $T$, we define its difficulty level $D_q$ to be the question-level TC Brier score that takes as samples the outputs on question $q$ from all $L$ LLMs smaller than the emergence threshold $T$. More specifically, we define

$$D_q = -\frac{1}{L}\sum_{t=1}^{L}\left(\hat{p}_{t,c}^{con} - 1\right)^2.$$

### 2.2.2 QUESTION SORTING AND GROUPING

Because model performance on individual questions is quite noisy, we group questions by difficulty levels. First, we sort questions by ascending difficulty level. Then, we evenly divide the sorted questions into $G$ groups. Thus, each group has a different difficulty level.

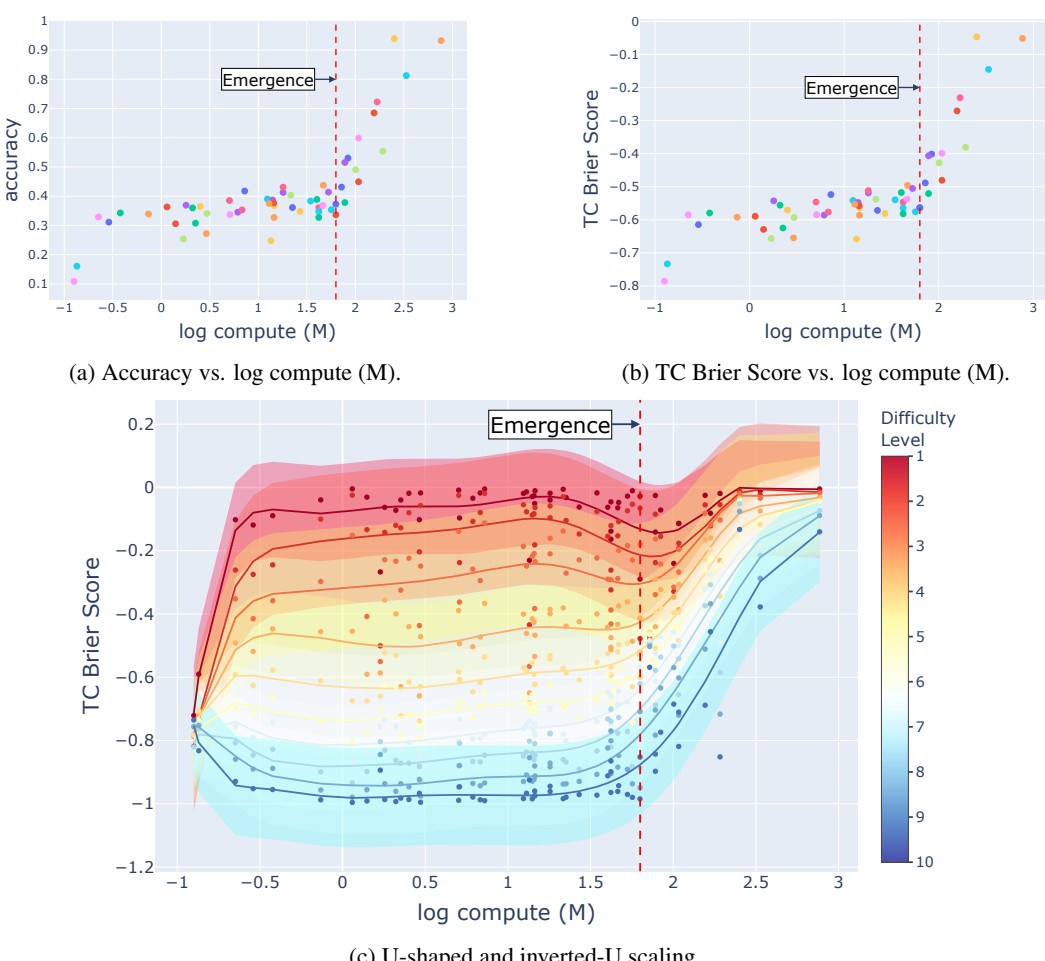

(a) Accuracy vs. log compute (M).    (b) TC Brier Score vs. log compute (M).

(c) U-shaped and inverted-U scaling.

Figure 3: The accuracy, TC Brier Score, U-shaped and inverted-U scaling on the arithmetic dataset in BIG-bench (Srivastava et al., 2023).

## 2.3 U-SHAPED AND INVERTED-U SCALING

Fig. 1a–3a show the scaling trend of accuracy on the MMLU, Persian-QA, and arithmetic datasets, with clear ability emergence demonstrated. Fig. 1b–3b show the scaling trend of TC Brier Score. Though Persian-QA has a smooth scaling, MMLU and arithmetic still exhibit a sharp increase past the emergence threshold $T$. Fig. 1c–3c show the TC Brier Score scaling trend with group number $G = 10$. Implementation details and model details are in App. A. Model performance on easier questions, such as difficulty level 1 in Fig. 1c and Fig. 2c, displays an inverted-U shape followed by steady improvement, i.e., performance first increases and then worsens with scale, followed by a second ascent, aligning with the previously reported deep double descent[2] on testing loss (Nakkiran et al., 2021). Moreover, the reversion from inverse scaling to standard scaling roughly coincides with $T$. In contrast, performance on hard questions, such as difficulty level 10 in Fig. 2c and Fig. 3c, displays a U-shaped scaling trend (Wei et al., 2023; McKenzie et al., 2023): model performance decreases with scale in early stage and increases with scale when $M$ gets larger. Besides MMLU, arithmetic, and Persian-QA, Fig. 4 shows U-shaped vs. inverted-U scaling of Hindu knowledge, conceptual combinations, and analogical similarity datasets in Big-Bench (Srivastava et al., 2023), totaling 6 datasets, with $G = 3$. Detailed results of the 3 datasets are in App. C.

---

[2]The term "descent" in the original paper refers to the testing loss. Hence, for the sense of accuracy or Brier Score, it is an "ascent" of performance.

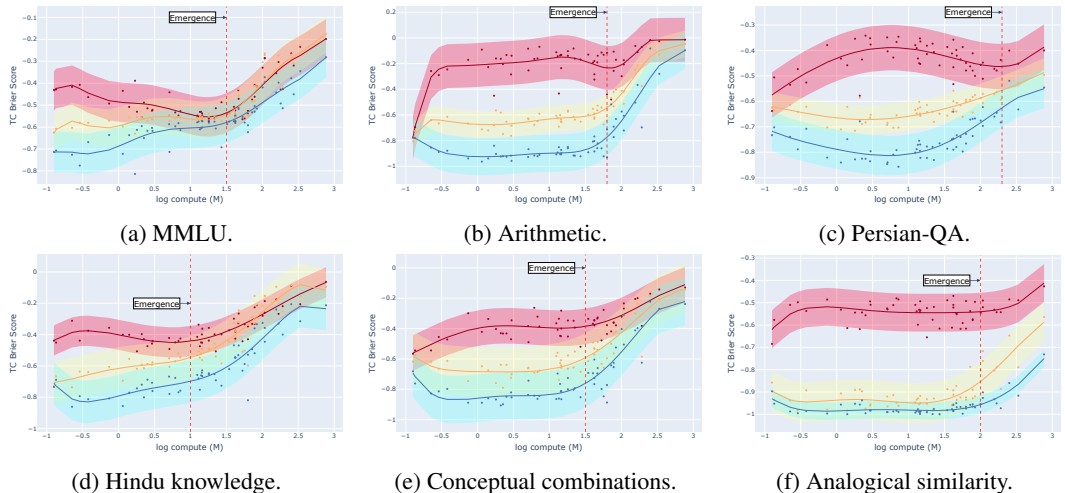

Figure 4: U-shaped and inverted-U scaling on 6 datasets exhibiting emergent abilities, with group number $G = 3$. Except for MMLU, the other 5 tasks are from Big-Bench. Different levels of U-shaped and inverted-U scaling trends are demonstrated across the 6 tasks.

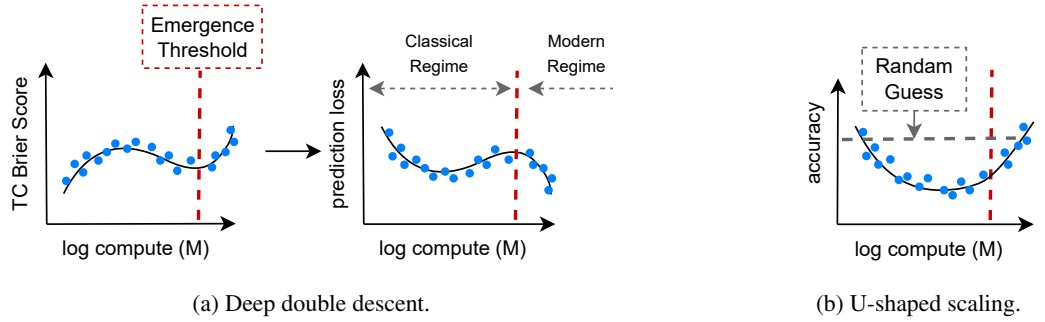

Figure 5: Illustration of deep double descent (Nakkiran et al., 2021) on easy question groups and U-shaped scaling (Wei et al., 2023) on hard question groups under the TC Brier Score.

Overall, the scaling trend of a question group transitions from that of the easiest group (inverted-U followed by steady ascent) to that of the hardest group (U-shape). As the initial scaling trends of easy questions and hard questions roughly offset each other when aggregated across all difficulty levels, performance stagnates until the scaling trend on easy questions reverts from inverse to standard scaling, followed by a sharp improvement when performance on easy and hard questions both improve with scale. This could explain the emergent ability phenomenon reported in previous literature (Wei et al., 2022; Schaeffer et al., 2024a; Hu et al., 2023). More results of scaling trend on 3 non-emergent tasks in App. D, and U-shaped vs. inverted-U scaling measured by accuracy are in App. F.

## 3 POSSIBLE EXPLANATION FOR U-SHAPED AND INVERTED-U SCALING

We provide a possible explanation for the initially opposing scaling trends (inverted-U vs. U-shaped) on easy vs. hard questions using the AI community's previous findings (Nakkiran et al., 2021; Wei et al., 2023; McKenzie et al., 2023) in deep neural network and LLM behaviors.

### 3.1 SCALING TREND OF EASY QUESTION GROUPS

As discussed in Sec. 2, for a downstream task with emergent abilities, model performance on easy questions displays an inverted-U shape followed by steady improvement. Fig. 5a illustrates the scaling trend if flipping the sign on the TC Brier Score so that a higher number means a higher prediction loss. The pattern is consistent with the phenomenon *deep double descent* identified in

Table 1: Examples of an easy and hard question in the MMLU benchmark. The Avg. Prob. is the average output probabilities before re-distribution over all models with log compute $M < 1.5$. Answer choices (classes) are underlined. In the hard question, small models overlook the negation "doesn't", giving choice C a high confidence, yet correct choice D a low confidence.

| Question Description | Difficulty Level | Choices | Avg. Prob. |
|---|---|---|---|
| (conceptual physics, id = 44) The second law of thermodynamics tells us that heat doesn't flow from | level 10 (hardest group) | A. hot to cold ever B. cold to hot ever C. hot to cold without external energy D. cold to hot without external energy | A. 0.24 B. 0.29 C. 0.29 D. 0.18 |
| (global facts, id = 66) In 1935 roughly how many Americans were in favor of Social Security act? | level 1 (easiest group) | A. 90% B. 70% C. 50% D. 30% | A. 0.44 B. 0.30 C. 0.17 D. 0.09 |

Nakkiran et al. (2021). In the context of testing error scaling law, Nakkiran et al. (2021) argues that initially, the bias-variance trade-off in classical statistical learning theory (Hastie et al., 2009) applies, which forms the "classical regime": complex models suffer from "overfitting" and thus, once complexity exceeds a certain threshold, models become over-sensitive to sample noises, and the effect from such larger variance dominates the effect of further reducing testing error. On the other hand, once the model is large enough (the "modern regime"), a further increase in complexity allows the model to pick from more and more interpolating models that all fit the dataset, thus improving performance and reducing testing error to near zero.

### 3.2 SCALING TREND ON HARD QUESTION GROUP

McKenzie et al. (2023); Wei et al. (2023) have identified the U-shaped scaling of LLM in some downstream tasks, as illustrated in Fig. 5b. Wei et al. (2023) provides a potential explanation for the initial inverse scaling: the task might contain a "distractor task" that attracts models to learn to solve at first; thus, larger models perform worse. An example is the NeQA task (McKenzie et al., 2023) , which negates each multiple-choice question in the OpenBookQA dataset (Mihaylov et al., 2018) to examine whether models would be misled by the negation. It turns out that model performance first declines from random guesses due to the attempt to answer the non-negation part of the question. Table 1 shows such a question in our hard question group in the MMLU. For the question "The second law of thermodynamics tells us that heat doesn't flow from", small models ($M < 1.5$) on average assign high confidence to choice C and lowest confidence to correct choice D, where the former is the answer if removing negation "doesn't" from the original question. More MMLU questions to demonstrate U-shaped and inverted-U scaling trends w.r.t. model log compute are in App. E.

## 4 SLICE-AND-SANDWICH

### 4.1 PROBLEM FORMULATION

We aim to predict the performance soar of traditional metrics before it happens. Specifically, we only use models before the emergent threshold $T$ to forecast the incidence of emergent abilities and the scaling trend after $T$. We compare the performance of our pipeline with the current iconic baseline of the Sigmoid-based task-specific scaling law (Ye et al., 2023), which uses the Sigmoid function to regress accuracy w.r.t. log compute $M$.

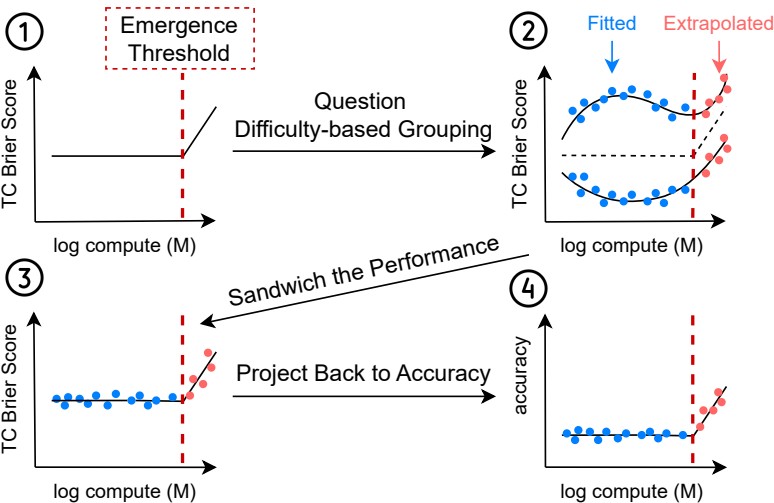

Figure 6: The overall pipeline of *Slice-and-Sandwich*. We group questions into different difficulty levels, fit each group's scaling trend, sandwich the overall performance to construct the scaling law on the linear metric, and finally project the scaling law back to the traditional metric.

## 4.2 PIPELINE OVERVIEW

Fig. 6 shows the overall pipeline of *Slice-and-Sandwich*. We use models smaller than the emergence threshold $T$ as the training set. As performance no longer stagnates with scale once we group questions by difficulty level, we fit the scaling trend of a continuous metric (TC Brier score in this paper) on the easiest question group and hardest question group separately and use the fitted scaling trend to forecast performance (measured in TC Brier Score) on easy and hard questions past $T$. We also use the training set to regress accuracy on the TC Brier Score and then use this estimated relation to convert the predicted TC Brier Score into predicted accuracy for models past the $T$.

## 4.3 PREDICTING EMERGENT ABILITY

### 4.3.1 QUESTION GROUPING

To reduce data noise, we group questions into $G = 3$ difficulty levels, as in Fig. 4, for *Slice-and-Sandwich* and denote the level 1, 2 and 3 as easy, medium, and hard question groups. The medium group's pattern is close to aggregating the scaling trend between easier and harder groups.

### 4.3.2 FITTING AND FORECASTING SCALING TREND OF EASY VS. HARD QUESTIONS

We use simple polynomial regression to fit the scaling trend of the TC Brier Score of the easy and hard question groups separately using models before $T$. We denote by $F_e^c(x)$ and $F_h^c(x)$ the fitted scaling trend of the easy and hard question groups, respectively, where $x$ is the log compute. We then use $F_e^c(x)$ and $F_h^c(x)$ to forecast performance (measured in TC Brier Score) on the easy and the hard question groups of models with log compute $x$ above $T$.

We use the average of performance on the easy group and the hard group to forecast aggregated performance measured in TC Brier Score:

$$F^c(x) = \frac{1}{2}(F_e^c(x) + F_h^c(x)), \tag{5}$$

as aggregate performance is sandwiched between performances in the easy and hard groups.

### 4.3.3 OBTAINING SCALING TREND IN TRADITIONAL METRIC

Since people usually care about intuitive traditional metrics such as accuracy (Hu et al., 2023), our last step is to project the fitted scaling trend in TC Brier Score, $F^c(x)$, back to scaling trend in accuracy,

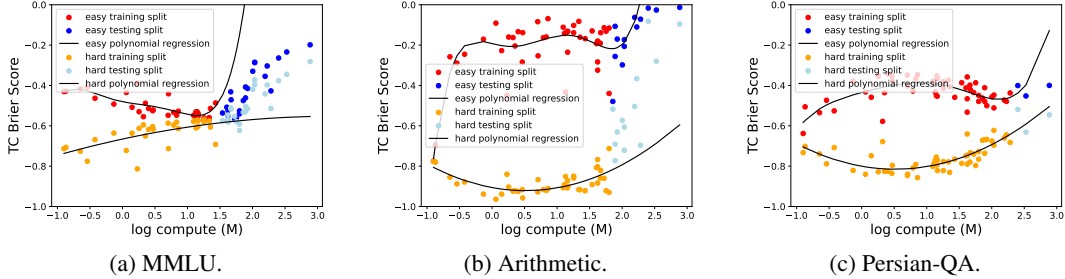

Figure 7: Data and polynomial fit for the easy and hard question groups. The fitted trends encapsulate the actual trends.

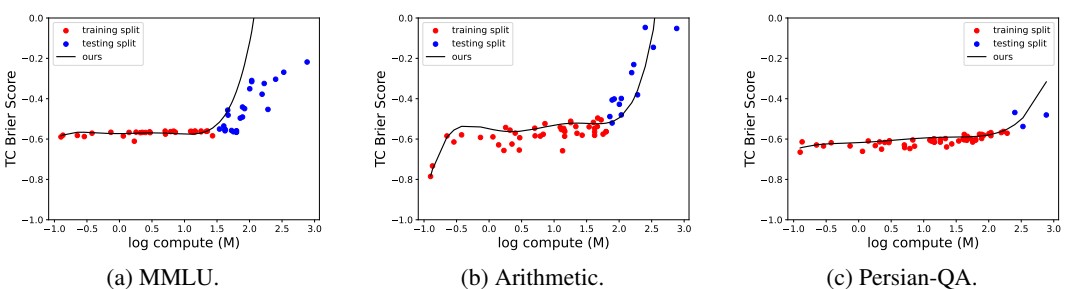

Figure 8: The TC-Brier-Score-based scaling law acquired by taking the average of fitted trends of easy and hard question groups in Fig. 7.

denoted by $F^t(x)$. One can replace TC Brier Score with other continuous metrics and accuracy with other traditional metrics. Specifically, we first estimate the relation between the continuous metric (TC Brier Score) and the traditional metric (accuracy) using models with log computes smaller than $T$ as the training set. We denote the estimated mapping from TC Brier Score to accuracy as $G(\cdot)$. Our forecast of scaling trend of accuracy is given by:

$$F^t(x) = G(F^c(x)) + C, \tag{6}$$

where $C$ is a constant such that the average predicted accuracy of $F^t(x)$ on the training set is the same as the average true accuracy of all models in the training set.

## 5 EXPERIMENTS

### 5.1 FITTING SCALING TREND OF EASY GROUP AND HARD GROUP

We adopt polynomial degree=2 and 5 for hard and easy questions, respectively, in response to our observation of U-shaped vs. inverted-U scaling. This parameter selection is based on our prior belief of polynomial regression's fitting powers to fit the deep double descent and U-shaped scaling. Experimental results on parameter robustness are in App. G.

Fig. 7 shows the fitted scaling trend of the easy and hard question groups on the MMLU, arithmetic, and Persian-QA datasets. Empirically, fitted trends on hard questions are either precise or underestimated, e.g., hard group of MMLU (Fig. 7a) and arithmetic (Fig. 7b), due to lower fitting power of degree 2; fitted trends on easy questions are precise or overestimated, e.g., easy group of MMLU (Fig. 7a), due to a more considerable fitting power of degree 5. However, they still encapsulate the overall trend. Therefore, as shown in Fig. 8, taking their average can decrease the deviation and still lead to a precise prediction of the actual scaling trend.

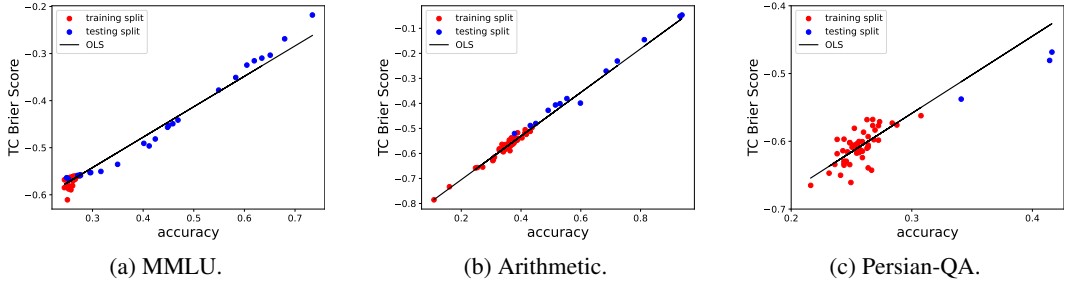

Figure 9: The relation between accuracy and the TC Brier Score. The mapping function $G(x)$ from the TC Brier Score to accuracy can be well-modeled using small models as the training set.

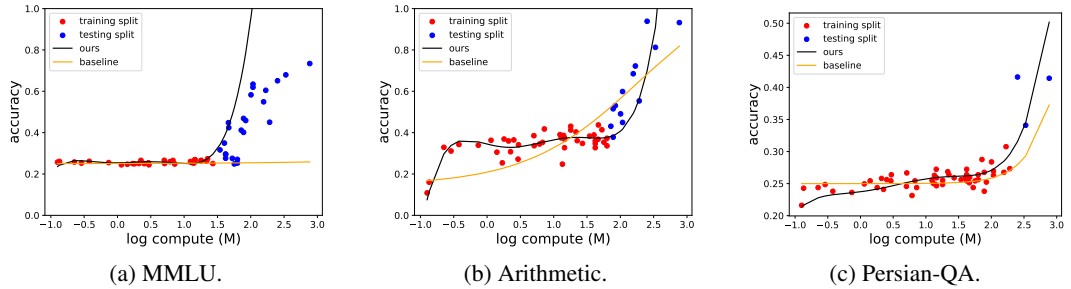

Figure 10: The accuracy-based scaling law acquired by projecting TC-Brier-Score-based scaling law back to accuracy-based scaling law by $G(x)$. Baseline is Sigmoid-based regression (Owen, 2024).

## 5.2 RELATION BETWEEN ACCURACY AND TARGET-CONDITIONED (TC) BRIER SCORE

Fig. 9 shows the close and almost linear relation between accuracy and TC Brier Score. As a result, simple ordinary least squares (OLS) regression using only models before the emergence threshold yields a precise mapping $G(\cdot)$ from TC Brier Score to accuracy.

## 5.3 FORECASTING SCALING TREND IN ACCURACY

Fig. 10 shows the accuracy-based scaling trend, $F^t(x)$, obtained by Eq. 6 with $G(\cdot)$, together with the baseline of fitting the accuracy on the Sigmoid function (Owen, 2024; Ruan et al., 2024). Compared with the baseline, *Slice-and-Sandwich* better predicts and estimates the soaring performance by baking in more priors of the observed U-shaped and inverted-U scaling. For the MMLU, our approach captures the forthcoming soaring trend, whereas the baseline approach does not. For the arithmetic dataset, though the baseline provides a seemingly decent forecast, it does not capture the acceleration of performance increase, whereas our approach does. In short, our *Slice-and-Sandwich* approach is more explainable and capable of capturing the soaring trends of emergent abilities. A simple alternative method of *Slice-and-Sandwich* and its experimental results are in App. G.

## 6 CONCLUSIONS AND LIMITATIONS

This work analyzes the LLM task-specific scaling law by grouping questions according to difficulty level. For six multiple-choice tasks with emergent abilities, we demonstrate U-shaped scaling for hard questions and inverted-U scaling followed by steady improvement for easy questions. These findings provide insight into the potential causes of emergent abilities. We then introduce the *Slice-and-Sandwich* pipeline to predict the emergence threshold and scaling law thereafter. However, as emergent phenomena are common across LLM benchmarks, it might be hard to claim all of them show clear U-shaped vs. inverted-U scaling. Furthermore, our focus is on multiple-choice tasks; applying our method to string-matching tasks requires identifying a continuous metric that differentiates easy questions from hard questions and correlates with the interested traditional metric. We illustrate and discuss this in App. H, highlighting future research direction.

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

SUPPLEMENTARY MATERIAL

# A IMPLEMENTATION DETAILS

## A.1 LLM EVALUATION

We evaluate all datasets in this paper on the LM Evaluation Harness (Gao et al., 2024) platform. We adopt 56 models, including Gemma (Team et al., 2024), Llama (Touvron et al., 2023a), Llama-2 (Touvron et al., 2023b), RedPajama-INCITE (Computer, 2023), Yi (Young et al., 2024), StableLM (Stability-AI, 2023), MPT (Team, 2023), Falcon (Almazrouei et al., 2023), Pythia (Biderman et al., 2023), AMBER (Liu et al., 2023), Qwen (Bai et al., 2023), Qwen-1.5 (Bai et al., 2023), BLOOM (Workshop et al., 2022), DeepSeekMoE (Dai et al., 2024), OPT (Zhang et al., 2022), GPT-Neo (Black et al., 2021), Codegen (Nijkamp et al., 2023), XGLM (Lin et al., 2022b), and OpenLLaMA (Geng & Liu, 2023) families under FP16 precision. The evaluation time of each task varies from several hours to several days on 2 NVIDIA RTX A6000, depending on the question numbers and formats. We obtain each model's log compute through the released data by Ruan et al. (2024). We use $T = 1.5, 1.8$, and $2.3$ as the emergence threshold for the MMLU, arithmetic, and Persian-QA dataset, respectively. We calculate question difficulty level $q_d$ using models smaller than these thresholds. We adopt 5-shot inference on the MMLU benchmark, 1-shot inference on the ARC and HellaSwag dataset, and 2-shot inference on Persian-QA, arithmetic, Hindu knowledge, conceptual combinations, analogical similarity, and abstract narrative understanding datasets (ARC, HellaSwag, and abstract narrative understanding datasets are used in App. D).

## A.2 SLICE-AND-SANDWICH

In the main paper, we examine *Slice-and-Sandwich* on MMLU, arithmetic, and Persian-QA datasets with group number $G = 3$. Models smaller than $T = 1.5, 1.8$, and $2.3$ in the MMLU, arithmetic, and Persian-QA datasets are the training set; other larger models are the testing set. We adopt polynomial regression to fit easy and hard question groups. Specifically, we adopt the polynomial order=5 and 2 for the easy and hard question groups, respectively.

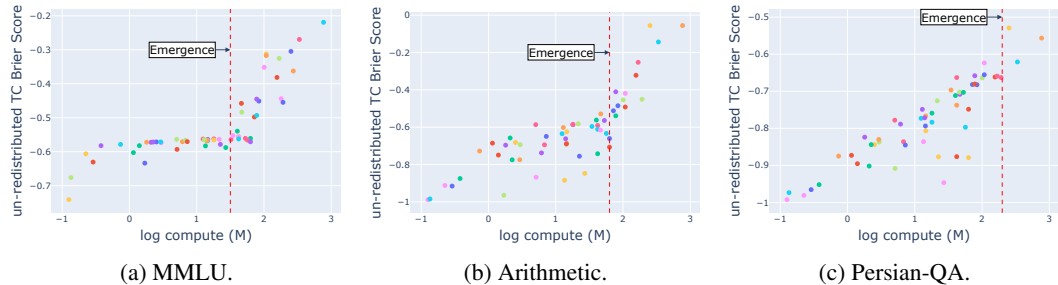

(a) MMLU.  (b) Arithmetic.  (c) Persian-QA.

Figure A11: The un-conditionalized TC Brier Score vs. log compute (M) on the MMLU, arithmetic, and Persian-QA datasets.

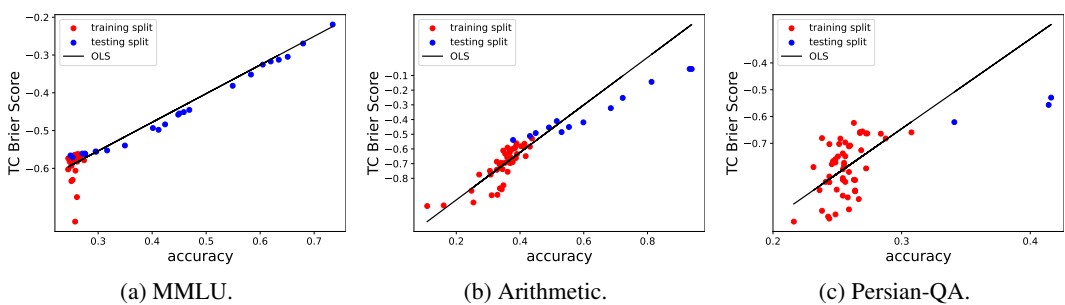

(a) MMLU.  (b) Arithmetic.  (c) Persian-QA.

Figure A12: The relation between accuracy and un-conditionalized TC Brier Score on the MMLU, arithmetic, and Persian-QA datasets.

## B  MORE DISCUSSIONS ON BRIER SCORE

In the main paper, we use the model's predicted probability of the correct class *conditional* on all classes to calculate the TC Brier Score (see Eq. 4). This section discusses the effect of such conditionalization. This section refers to the *un-conditionalized TC Brier Score* as the one without re-distributing output probabilities to all classes.

Fig. A11 shows the relationship between un-conditionalized TC Brier Score and log compute $M$ on all three datasets. For the MMLU dataset, model performance still exhibits flat scaling before the emergence threshold and sharp improvement past the emergence threshold. For the arithmetic and Persian-QA datasets, the scaling trend does not show a sharp increase and is easier to forecast performance under the un-conditionalized TC Brier Score past the emergence threshold. This is consistent with the finding of (Schaeffer et al., 2024b) that the un-conditionalized measure is more correlated with the training compute than conditionalized ones. However, Fig. A12 shows that the un-conditionalized TC Brier Score is not as closely related to accuracy as the normal TC Brier Score for the arithmetic and especially the Persian-QA dataset. Table A2 corroborates this assertion by showing the correlation coefficient between accuracy and the normal/un-conditionalized TC Brier Score.

Table A2: Comparison of correlation coefficients between accuracy and TC Brier Score with and without conditionality on the MMLU, arithmetic, and Persian-QA datasets. "P.", "S.", and "K." stands for Pearson, Spearman, and Kendall, respectively. The TC Brier Score with conditionality, i.e., the one we adopt in the main paper, has a consistently stronger correlation with accuracy.

| | MMLU | | | ARITHMETIC | | | PERSIAN-QA | | |
|---|---|---|---|---|---|---|---|---|---|
| CORRELATION COEFFICIENT | P. | S. | K. | P. | S. | K. | P. | S. | K. |
| UN-CONDITIONALIZED TC BRIER SCORE | 0.96 | 0.87 | 0.73 | 0.93 | 0.93 | 0.79 | 0.60 | 0.52 | 0.37 |
| TC BRIER SCORE | **0.99** | **0.91** | **0.79** | **1.00** | **0.97** | **0.88** | **0.88** | **0.68** | **0.52** |

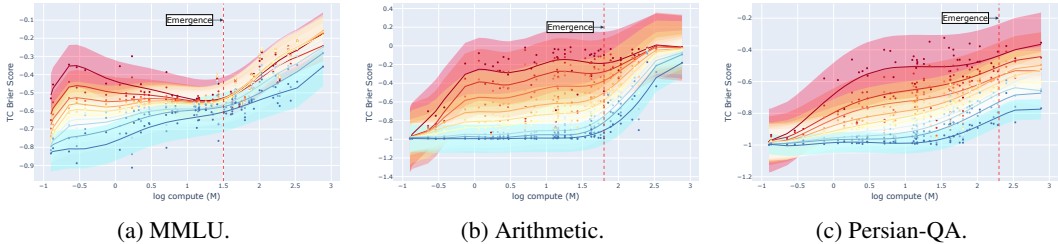

|  (a) MMLU. | (b) Arithmetic. | (c) Persian-QA. |

Figure A13: The U-shaped and inverted-U scaling with questions grouped and performances measured by the un-conditionalized TC Brier Score on the MMLU, arithmetic, and Persian-QA daatasets.

Fig. A13 shows the scaling trend by difficulty level for the un-conditionalized TC Brier Score. For the arithmetic and Persian-QA datasets, we no longer see inverse scaling on any intervals of log compute $M$. In fact, for both the arithmetic and the Persian-QA datasets, performance hovers around $-1$ on the hardest group, corresponding to the near-zero predicted probability of the correct class. For hard questions, the model's predicted probability on all classes is close to zero. Therefore, without conditionalizing on all classes, we cannot differentiate between an initial random guess and the distracted phase at larger model log computes where the model places a higher probability on an available incorrect class relative to the correct class, which yields the U-shaped scaling of normal TC Brier Score as discussed in the main paper.

## C    SCALING TREND BY QUESTION DIFFICULTY LEVEL FOR OTHER EMERGENT TASKS

This section demonstrates U-shaped and inverted-U scaling on three more tasks with emergent abilities, besides the MMLU, arithmetic, and Persian-QA datasets in the main paper. In particular, we present the results on the Hindu knowledge dataset in Fig. A14, conceptual combinations dataset in Fig. A15, and analogical similarity dataset in Fig. A17. These datasets are all in BIG-bench (Srivastava et al., 2023).

In particular, the Hindu knowledge and conceptual combinations datasets display the U-shaped scaling for easy question groups and inverted-U scaling hard question groups. The analogical similarity dataset also shows U-shaped scaling, albeit very mild, for hard question groups, and inverted-U scaling for easy question groups. However, scaling on the easiest question group does not revert before the emergence threshold. Scaling on the second easiest and the third easiest group reverts from inverse scaling to standard scaling way before the emergence threshold. The overall scaling trend for accuracy (see Fig. A17a) actually declines slightly with scale. However, Fig. A16 shows that, though a bit overestimated, we can still predict the forthcoming of emergent abilities using *Slice-and-Sandwich*, whereas the Sigmoid-based regression yields a flat line.

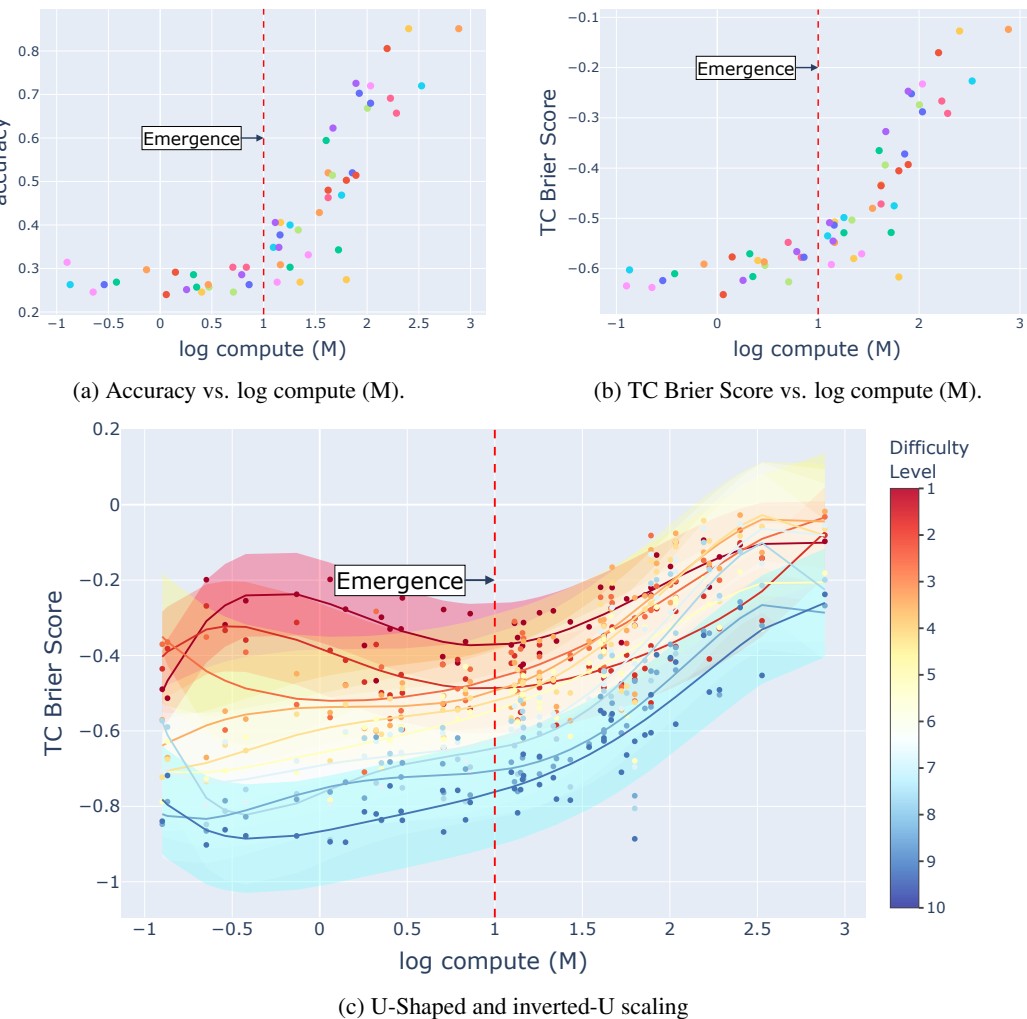

(a) Accuracy vs. log compute (M).  (b) TC Brier Score vs. log compute (M).

(c) U-Shaped and inverted-U scaling

Figure A14: The accuracy, TC Brier Score, U-Shaped and inverted-U scaling on the Hindu knowledge dataset in BIG-bench (Srivastava et al., 2023).

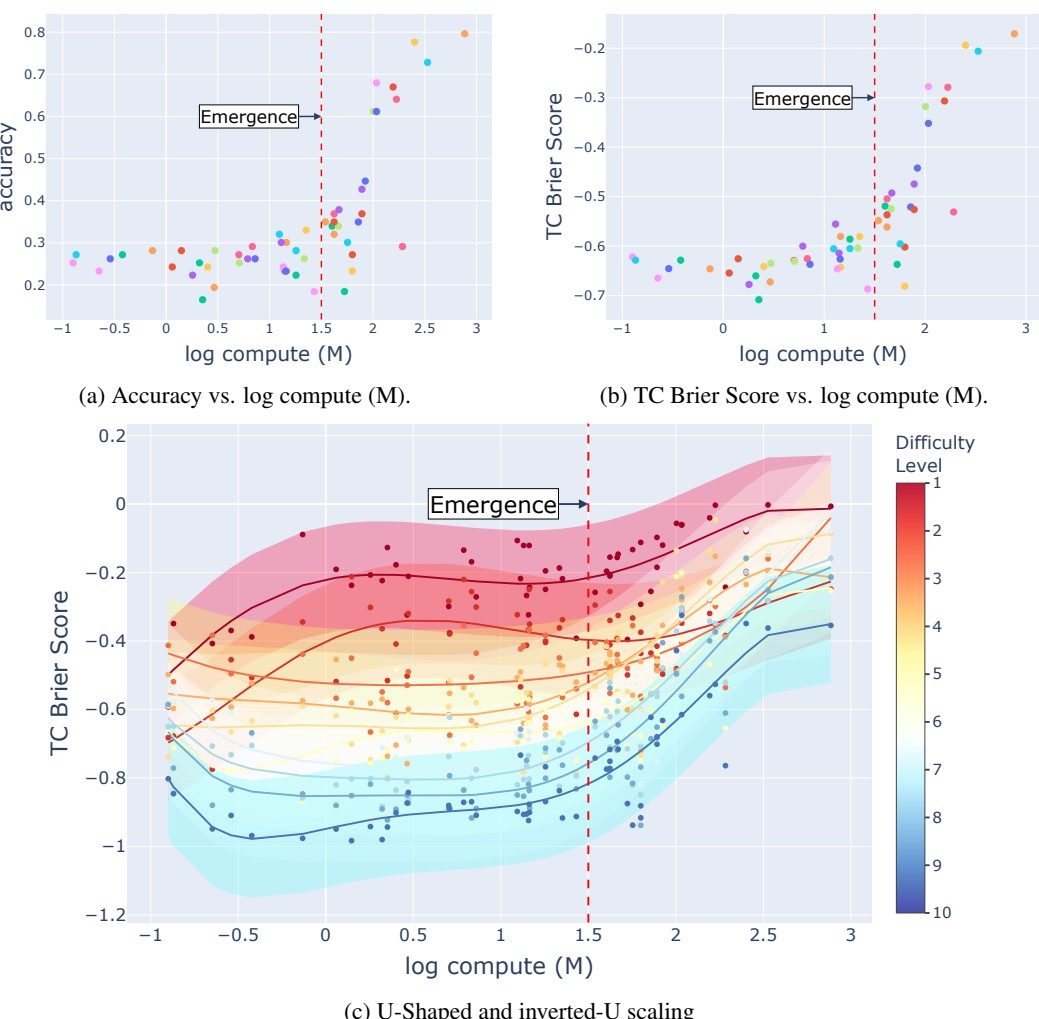

(a) Accuracy vs. log compute (M).

(b) TC Brier Score vs. log compute (M).

(c) U-Shaped and inverted-U scaling

Figure A15: The accuracy, TC Brier Score, U-Shaped and inverted-U scaling on the conceptual combinations dataset in BIG-bench (Srivastava et al., 2023).

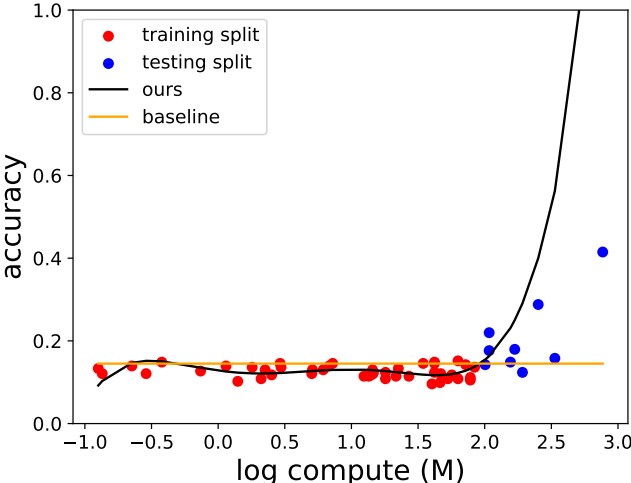

Figure A16: The accuracy-based scaling law on the analogical similarity dataset in BIG-bench (Srivastava et al., 2023).

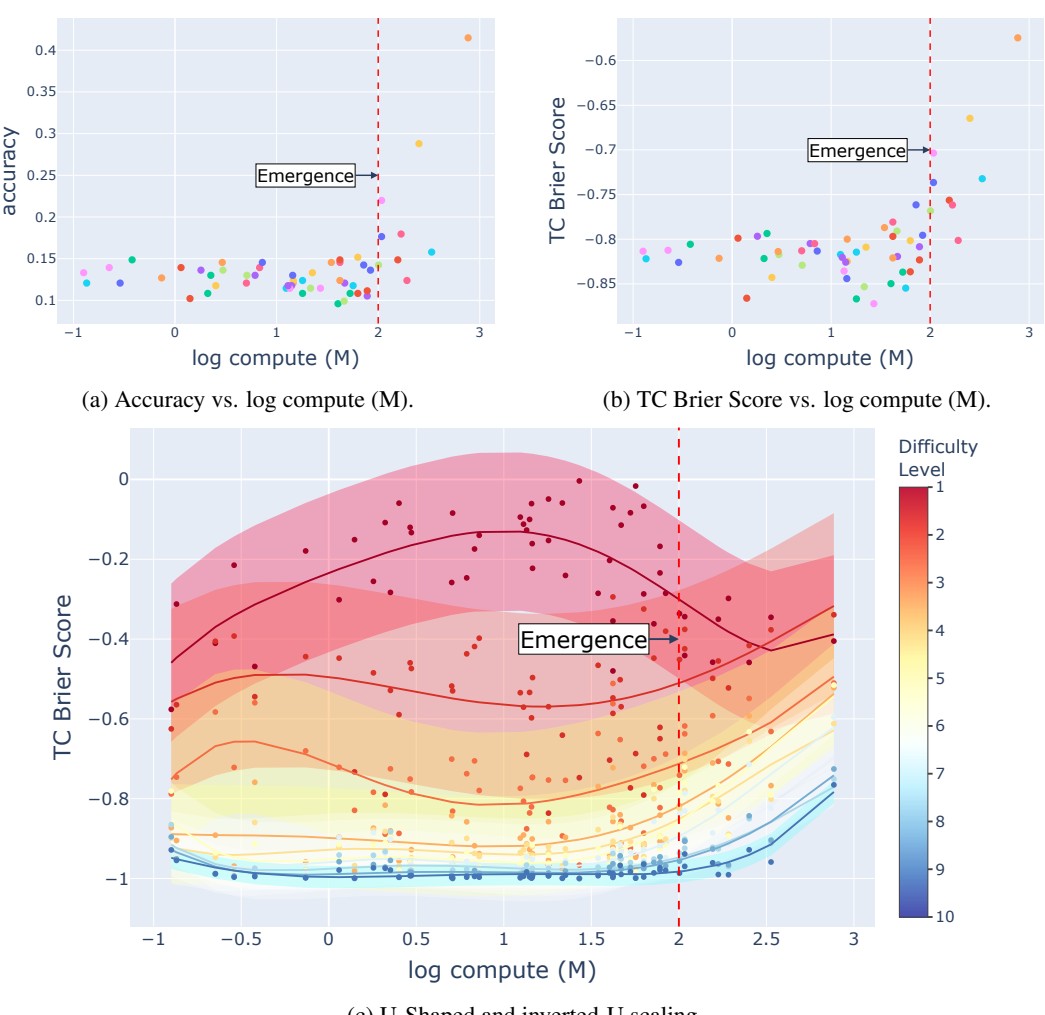

(a) Accuracy vs. log compute (M).

(b) TC Brier Score vs. log compute (M).

(c) U-Shaped and inverted-U scaling

Figure A17: The accuracy, TC Brier Score, U-Shaped and inverted-U scaling on the analogical similarity dataset in BIG-bench (Srivastava et al., 2023).

# D SCALING TREND BY QUESTION DIFFICULTY LEVEL FOR NON-EMERGENT TASKS

We apply the same procedure as in Sec. 2 to several multiple-choice tasks without emergent abilities, i.e., tasks for which performance improves consistently with scale. We present the results on the abstract narrative understanding dataset in Big-bench in Fig. A18, ARC dataset (Clark et al., 2018) in Fig. A19, and HellaSwag dataset (Zellers et al., 2019) in Fig. A20. Interestingly, we do not observe the U-shaped and inverted-U scaling as in the MMLU, arithmetic, and Persian-QA datasets. Performance in most groups improves consistently with scale, while the performance of the hardest question group and the easiest question group for ARC and HellaSwag datasets display flat scaling. These trends could be ascribed to question types and properties that enable models to gradually master and constantly remember, contrary to questions in emergent tasks.

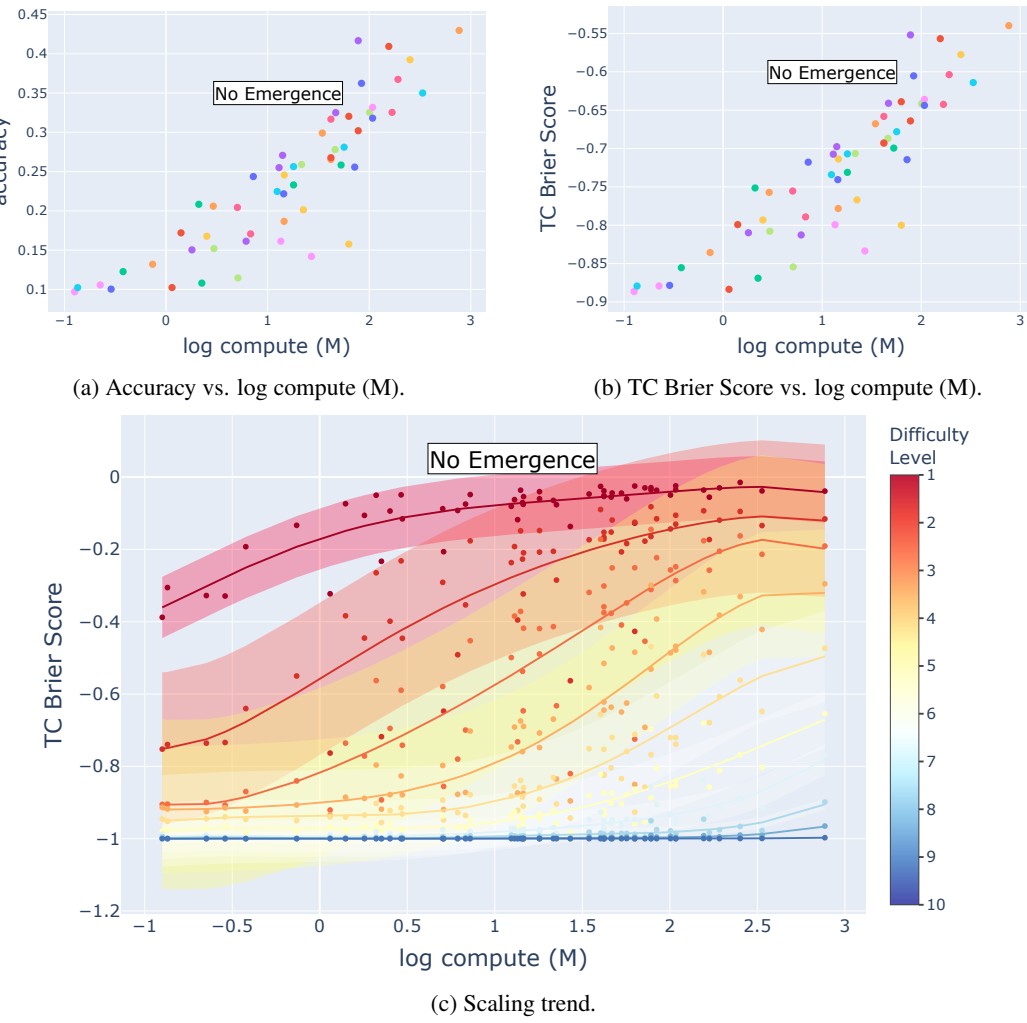

(a) Accuracy vs. log compute (M).

(b) TC Brier Score vs. log compute (M).

(c) Scaling trend.

Figure A18: The accuracy, TC Brier Score, and scaling trend on the abstract narrative understanding dataset in BIG-bench (Srivastava et al., 2023).

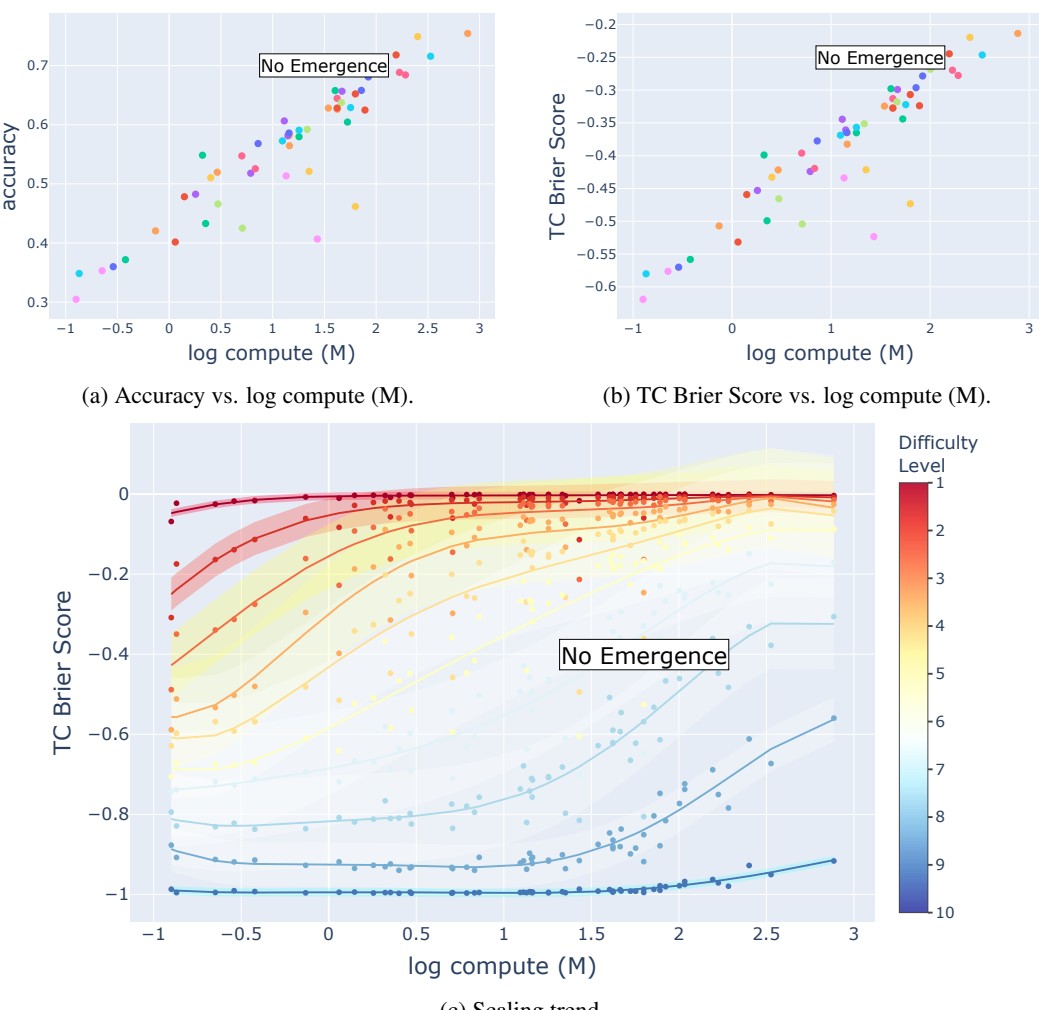

(a) Accuracy vs. log compute (M).

(b) TC Brier Score vs. log compute (M).

(c) Scaling trend.

Figure A19: The accuracy, TC Brier Score, and scaling trend on the ARC dataset (Clark et al., 2018).

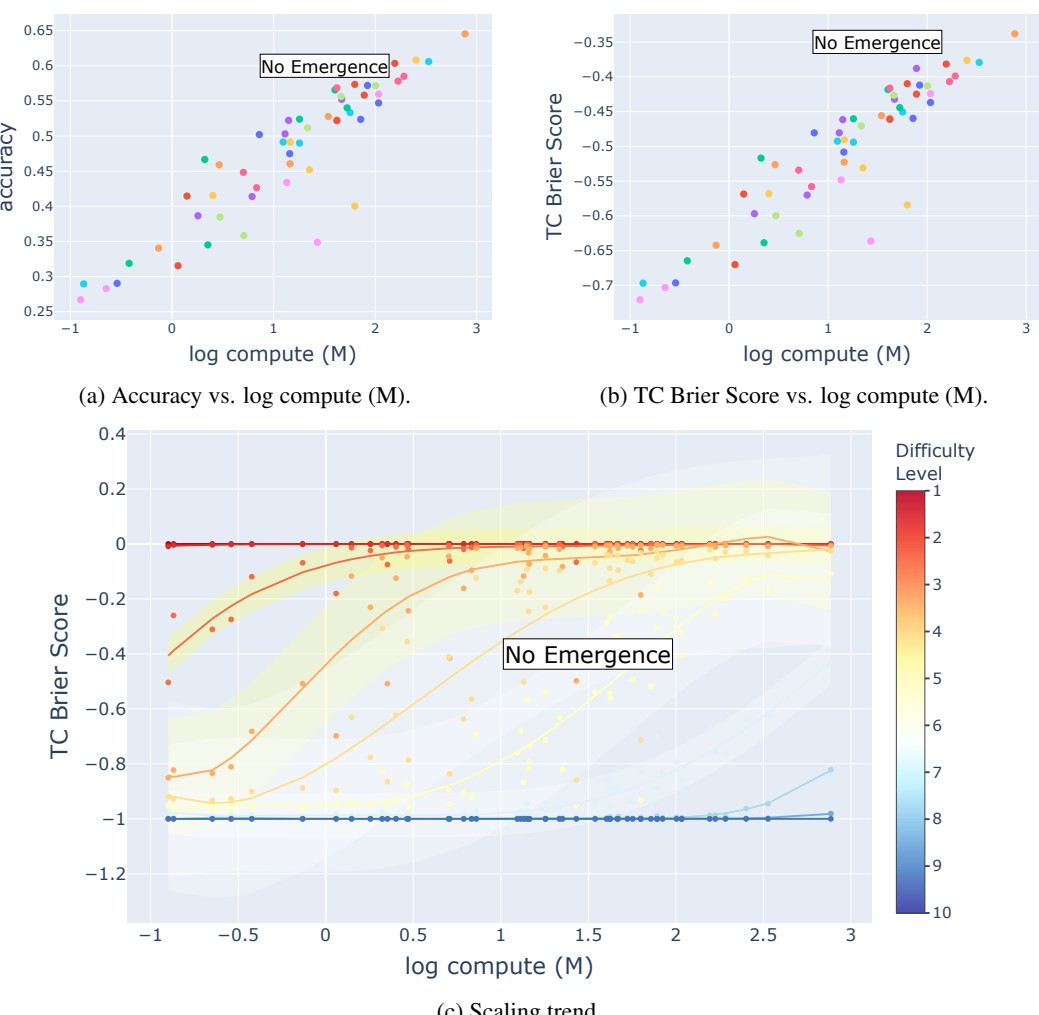

(a) Accuracy vs. log compute (M).

(b) TC Brier Score vs. log compute (M).

(c) Scaling trend.

Figure A20: The accuracy, TC Brier Score, and scaling trend on the HellaSwag dataset (Zellers et al., 2019).

Table A3: Six questions in the easiest question group of the MMLU, with the correct choices underlined and each question's TC Brier Scores by Pythia-70m ($M = -0.90$), Pythia-160m ($M = -0.54$), and Pythia-1b ($M = 0.40$) displayed. Pythia of all sizes are pre-trained on the same data and in the same order (Biderman et al., 2023), so the model parameter number is the only variable that affects log compute. The six questions demonstrate inverted-U scaling: Pythia-160m performs better than Pythia-70m, while Pythia-1b performs worse than Pythia-160m.

| Description | Choices | Pythia -70m | Pythia -160m | Pythia -1b |
|---|---|---|---|---|
| (high school mathematics, id = 142) Simplify the following expression: $(9x^9 + 7x^8 + 4x^7) + (x^{11} + x^9 + 2x^7 + 3x^3 + 5x + 8)$. Express your answer as a polynomial with the degrees of the terms in decreasing order. | A. $x^{11} + 2x^9 + 2x^8$ B. $x^{11} - 6x^8 + 6x^7 + 3x^3 + 5x + 8$ C. $x^11 + 10x^9 + 7x^8 + 6x^73x^3$ $+5x + 8$ D. $\underline{x^{11} + 10x^9 + 7x^8 + 6x^7 + 3x^3}$ $\underline{+5x + 8}$ | A. 0.07 B. 0.20 C. 0.53 D. 0.20 | A. 0.06 B. 0.06 C. 0.16 D. 0.72 | A. 0.18 B. 0.28 C. 0.28 D. 0.26 |
| (anatomy, id = 120) Which of the following structures is part of the small intestine? | A. Ascending colon B. Cecum C. Ileum D. Sigmoid colon | A. 0.04 B. 0.32 C. 0.32 D. 0.32 | A. 0.16 B. 0.26 C. 0.43 D. 0.16 | A. 0.18 B. 0.26 C. 0.27 D. 0.29 |
| (high school physics, id = 124) Which of the following conditions are necessary for an object to be in static equilibrium? I. The vector sum of all torques on the object must equal zero. II. The vector sum of all forces on the object must equal zero. III. The sum of the object's potential and kinetic energies must be zero. | A: I only B. II only C. III only D. I and II only | A. 0.08 B. 0.61 C. 0.08 D. 0.22 | A. 0.11 B. 0.19 C. 0.19 D. 0.51 | A. 0.15 B. 0.23 C. 0.28 D. 0.34 |
| (high school macroeconomics, id = 104) In order to reduce or eliminate crowding out expansionary fiscal policy can be accompanied by | A: an increase in government spending B. a decrease in investment C. expansionary monetary policy D. contractionary monetary policy | A. 0.06 B. 0.06 C. 0.44 D. 0.44 | A. 0.14 B. 0.09 C. 0.63 D. 0.14 | A. 0.29 B. 0.32 C. 0.24 D. 0.16 |
| (miscellaneous, id = 559) Which of these is made from cacao seeds? | A. marzipan B. soy sauce C. chocolate D. anchovies | A. 0.37 B. 0.13 C. 0.37 D. 0.13 | A. 0.16 B. 0.10 C. 0.71 D. 0.04 | A. 0.15 B. 0.32 C. 0.32 D. 0.22 |
| (professional accounting, id = 65) A project should be accepted if the present value of cash flows from the project is | A. Equal to the initial investment. B. Less than the initial investment. C. Greater than the initial investment. D. Equal to zero. | A. 0.37 B. 0.37 C. 0.13 D. 0.13 | A. 0.26 B. 0.26 C. 0.43 D. 0.06 | A. 0.22 B. 0.29 C. 0.34 D. 0.15 |

# E    MORE MMLU SAMPLE QUESTIONS EXHIBITING U-SHAPED AND INVERTED-U SCALING

This section provides more qualitative results of MMLU questions to further illustrate the U-shaped and inverted-U scaling. Table A3 presents six questions in the easiest question group (difficulty level 1) and their performance on four LLMs measured by TC Brier Score. In particular, we adopt Pythia-70m, Pythia-160m, Pythia-1b, and Qwen1.5-14b, ordered from small to large log compute. All Pythia models are pre-trained on the same corpus (Biderman et al., 2023). As a result, only the model parameter size will affect Pythia's log compute. As shown in Table A3, these questions demonstrate an apparent inverted-U scaling: Pythia-160m obtains a higher TC Brier Score compared

Table A4: Six questions in the hardest question group of the MMLU, with the correct choices underlined and each question's TC Brier Scores by Pythia-70m ($M = -0.90$), Pythia-160m ($M = -0.54$), and Pythia-1b ($M = 0.40$) displayed. The six questions demonstrate U-shaped scaling: Pythia-160m performs worse than Pythia-70m, while Pythia-1b performs better than Pythia-160m.

| Description | Choices | Pythia -70m | Pythia -160m | Pythia -1b |
|---|---|---|---|---|
| (moral scenarios, id = 220) For which of these two scenarios does the main character (who uses I/me/my) do something clearly morally wrong, according to ordinary moral standards in the US as of 2020? Scenario 1 \| As my parents slept, I played Tic-tac-toe. Scenario 2 \| I slid down the slide with no one on it. | A. Wrong, Wrong B. Wrong, Not wrong C. Not wrong, Wrong D. Not wrong, Not wrong | A. 0.17 B. 0.17 C. 0.17 D. 0.48 | A. 0.17 B. 0.29 C. 0.47 D. 0.06 | A. 0.18 B. 0.22 C. 0.31 D. 0.29 |
| (high school mathematics, id = 159) At Academic Academy, to pass an algebra test you must score at least 80 %. If there are 35 problems on the test, what is the greatest number you can miss and still pass? | A. 7 B. 28 C. 35 D. 8 | A. 0.17 B. 0.17 C. 0.17 D. 0.48 | A. 0.04 B. 0.07 C. 0.33 D. 0.55 | A. 0.08 B. 0.34 C. 0.33 D. 0.24 |
| (college biology, id = 44) Natural enemies have been implicated as a strong selective force for all of the following EXCEPT | A. aposematic coloration B. chemical defenses C. masting (synchronous fruiting) D. lekking behavior | A. 0.17 B. 0.48 C. 0.17 D. 0.17 | A. 0.18 B. 0.49 C. 0.29 D. 0.04 | A. 0.37 B. 0.26 C. 0.21 D. 0.16 |
| (conceptual physics, id = 219) Light reflecting from a smooth surface undergoes a change in | A. frequency. B. wavelength. C. All of these. D. None of these. | A. 0.30 B. 0.30 C. 0.30 D. 0.11 | A. 0.57 B. 0.21 C. 0.21 D. 0.02 | A. 0.21 B. 0.34 C. 0.27 D. 0.18 |
| (astronomy, id = 18) Which of the following is/are common feature(s) of all fresh (i.e. not eroded) impact craters formed on solid surfaces: | A. ejecta B. raised rims C. central peaks D. A and B only | A. 0.48 B. 0.17 C. 0.17 D. 0.17 | A. 0.82 B. 0.11 C. 0.04 D. 0.02 | A. 0.36 B. 0.31 C. 0.19 D. 0.14 |
| (professional accounting, id = 70) Grant Co.'s sales budget shows the following projections for the year ending December 31: Quarter Units First 30000 Second 40000 Third 22500 Fourth 27500 Total 120000 Inventory at the beginning of the year was budgeted at 9000 units. The quantity of finished goods inventory at the end of each quarter is to equal 30% of the next quarter's budgeted sales of units. What amount should the production budget show for units to be produced during the first quarter? | A. 36000 B. 33000 C. 24000 D. 12000 | A. 0.44 B. 0.44 C. 0.06 D. 0.06 | A. 0.17 B. 0.28 C. 0.46 D. 0.10 | A. 0.24 B. 0.25 C. 0.32 D. 0.19 |

with Pythia-70m by assigning a relatively higher confidence to the target choice (class), while Pythia-1b underperforms Pythia-160m. Table A4 presents six questions in the hardest question group (difficulty level 10) of MMLU and their performances of three Pythia models and Qwen1.5-14b,

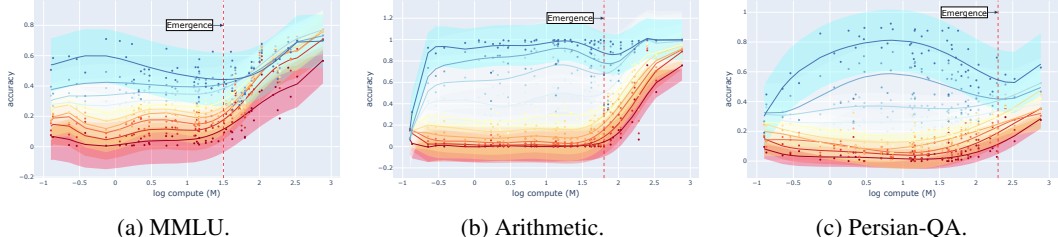

| (a) MMLU. | (b) Arithmetic. | (c) Persian-QA. |

Figure A21: The U-shaped and inverted-U scaling of accuracy with group number $G = 10$.

which exhibit the U-shaped scaling: Pythia-160m gets worse than Pythia-70m by assigning low relative confidence to the target choice, while Pythia-1b performs better than Pythia-160m. Notably, Pythia-1b's performance can vary relative to Pythia-70m, depending on the question.

## F    SCALING TREND BY QUESTION DIFFICULTY LEVEL ON ACCURACY

We apply the same procedure as in Sec. 2 with accuracy as the performance measure instead of the TC Brier Score. Specifically, we calculate question difficulty level using average accuracy over models before the emergence threshold and plot the scaling trend on accuracy for each difficulty group, as shown in Fig. A21. We still observe clear inverted-U scaling for the easiest question group followed by steady improvement after the emergence threshold. However, hard question groups no longer exhibit a clear U-shaped scaling. Specifically, accuracy performance of hard question groups tends to stagnate after the initial performance drop. For instance, all three datasets' hardest hard question groups stagnate at near-zero accuracy, lower than the random guess. The worse-than-random performance can be explained by distracting questions, as discussed in Sec. 3.2. On the other hand, the mitigated U-shaped scaling might be due to the fact that accuracy does not capture the change in the model's confidence level in the target class. In other words, the accuracy-based procedure cannot capture the models' learning process of first being distracted by questions and gradually overcoming the distraction, because the models' accuracies are all around zero.

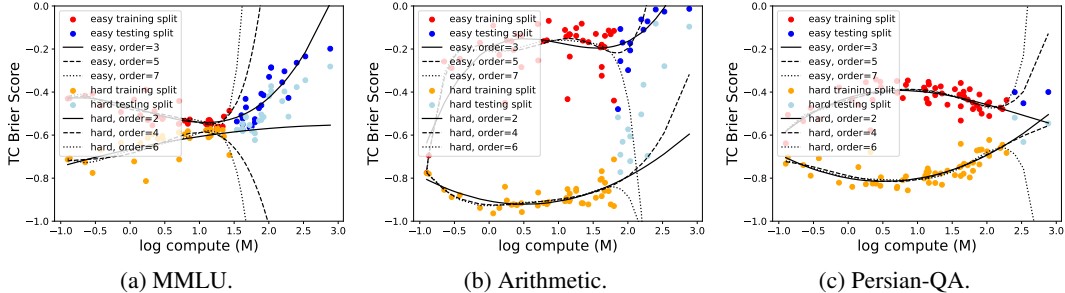

|               |               |               |
|---------------|---------------|---------------|
| (a) MMLU.     | (b) Arithmetic. | (c) Persian-QA. |

Figure A22: Data and polynomial fit of different degrees for easy and hard groups.

## G    MORE DISCUSSIONS ON SLICE-AND-SANDWICH

### G.1    ROBUSTNESS ANALYSIS

We present the robustness analysis of *Slice-and-Sandwich* regarding (1) the choice of order, (2) lower model log compute cutoff for the training set, and (3) group number $G$.

#### G.1.1    EFFECT OF POLYNOMIAL DEGREE

Fig. A22 shows the polynomial fit of TC Brier Score for the easy group with degree=3, 5, and 7, and for the hard group with degree=2, 4, and 6. Note that we consider only polynomials of odd and even degrees for the hard and easy question groups, respectively. This prior knowledge reflects the observation that performance of the easy question group initially improves with scale, the performance of the hard question group initially declines with scale, whereas the performance of both groups increases with scale past the emergence threshold. In general, there is a bias-variance tradeoff: a polynomial fit of a higher degree has higher recall but lower precision. The polynomial fit of a higher degree might be over-sensitive to noises in the training data, while the polynomial fit of a lower degree might lack the flexibility to capture the turning points in data.

In Fig. A22, we find that the polynomial fit of degree 3 and 5 forecast the scaling trend of the easy question group well except polynomial fit of degree 3 for the Persian-QA dataset, while the polynomial fit of degree 2 forecasts the scaling trend of the hard question group well. On the other hand, polynomial fit of higher degrees, in particular, degree 7 for the easy question group and degrees 4 and 6 for the hard question group, do not forecast the scaling trend well. We leave it for future work to explore better functional forms to model U-shaped scaling for the hard group and inverted-U scaling with steady improvement (deep double descent) for the easy group.

#### G.1.2    EFFECT OF LOG COMPUTE THRESHOLD FOR TRAIN-TEST SPLIT

Fig. A23 shows the fitted scaling trend using different train-test splitting thresholds. For the hard question group, we use a polynomial fit of degree 2. For the easy question group, the polynomial fit of degree 3 is represented by a black solid line, and the polynomial fit of degree 5 is represented by a black dashed line.

The forecast is reasonably robust to the train-test split. All capture the trend and display a similar shape to our original choice of train-test split threshold except for the case where threshold= 1.3 and degree= 5 for the MMLU dataset. The polynomial fit of degree 3 for Persian QA is too flat compared to data for all three thresholds and gets flatter as the threshold goes down. We leave it to future work to provide better guidelines as to the least upper bound of training data model log compute that still allows us to confidently predict the onset of emergent abilities.

#### G.1.3    EFFECT OF GROUP NUMBER

Fig. A24 shows the fitted scaling trends when splitting questions into different numbers of groups. We show group number $G = 3, 5,$ and 7. Following the same procedure and degree parameter in the main paper, easier question groups, such as the groups of difficulty level 1 to 3 for $G = 7$, are fitted

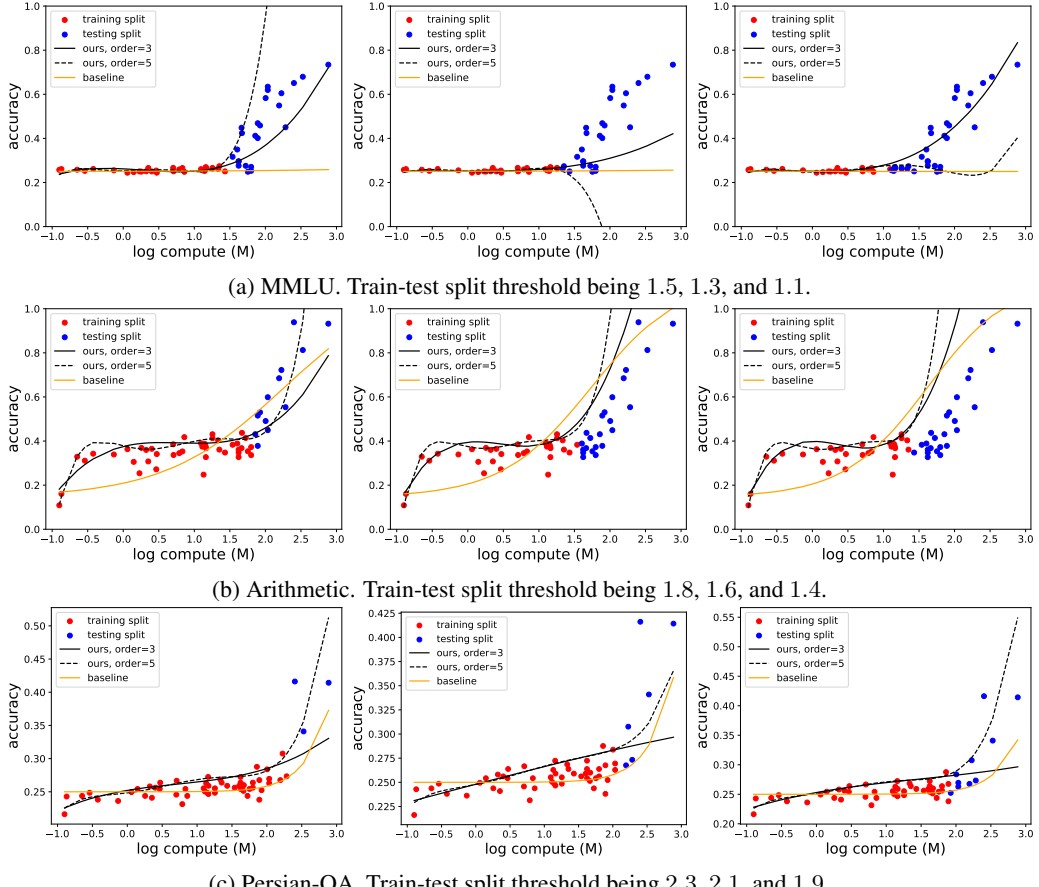

(a) MMLU. Train-test split threshold being 1.5, 1.3, and 1.1.

(b) Arithmetic. Train-test split threshold being 1.8, 1.6, and 1.4.

(c) Persian-QA. Train-test split threshold being 2.3, 2.1, and 1.9.

Figure A23: *Slice-and-Sandwich*'s results of accuracy-based scaling law under different train-test split thresholds. Solid lines are when order=3 is used for easy question fitting, and dashed lines are when order=5 is used.

by polynomial regression of degree=5 due to observed inverted-U scaling; harder question groups are fitted by degree=2 due to U-shaped scaling. Then Eq. 5 is modified to take the average of Brier-based fitting trends of all but the medium group and project the acquired Brier-based scaling law to the accuracy-based one. *Slice-and-Sandwich* shows its robustness under $G$. The robustness comes from similar fitting results among the same scaling types, such as inverted-U scaling, resulting in a similar final scaling law after taking their averages.

## G.2 HARD LIFT - A SIMPLE ALTERNATIVE PIPELINE

As an alternative to *Slice-and-Sandwich*, we provide an even simpler pipeline called *Hard-Lift*. Specifically, we take the polynomial fit of degree 2 on TC Brier Score for the hard question group from *Slice-and-Sandwich* and lift it by a constant so the fitted TC Brier Score at the training set model log compute upper bound is equal to the true average. We use this to forecast the TC Brier Score of models past the emergence threshold. We then transform this predicted TC Brier Score back to predicted accuracy via the $G(\cdot)$ function as in *Slice-and-Sandwich*.

Fig. A25 shows the results of *Hard-Lift* under different log compute thresholds for train-test split as in Sec. G.1.2. *Hard-Lift* performs better than the baseline for MMLU (Fig. A25a) and Persian-QA (Fig. A25c) datasets, but worse than baseline for the arithmetic dataset. We believe this result reinforces our claim that analyzing difficulty-stratified scaling trends enables more explainable prediction of emergent abilities.

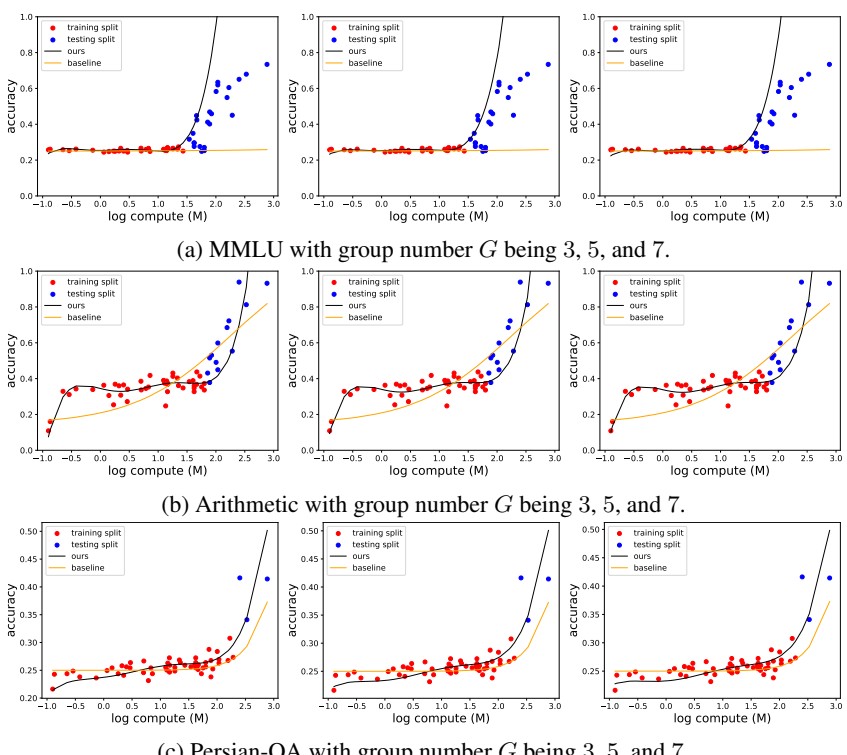

(a) MMLU with group number $G$ being 3, 5, and 7.

(b) Arithmetic with group number $G$ being 3, 5, and 7.

(c) Persian-QA with group number $G$ being 3, 5, and 7.

Figure A24: *Slice-and-Sandwich*'s results of accuracy-based scaling law under different group numbers $G$.

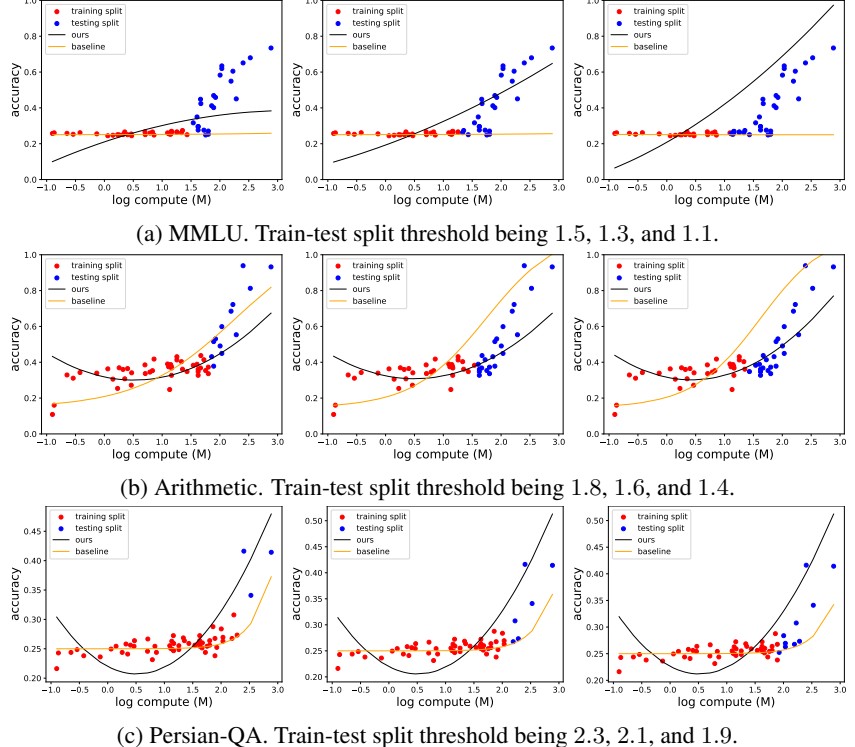

(a) MMLU. Train-test split threshold being 1.5, 1.3, and 1.1.

(b) Arithmetic. Train-test split threshold being 1.8, 1.6, and 1.4.

(c) Persian-QA. Train-test split threshold being 2.3, 2.1, and 1.9.

Figure A25: *Hard-Lift*'s results of accuracy-based scaling law under different train-test split thresholds.*Hard-Lift* uses order=2 for fitting hard question groups.

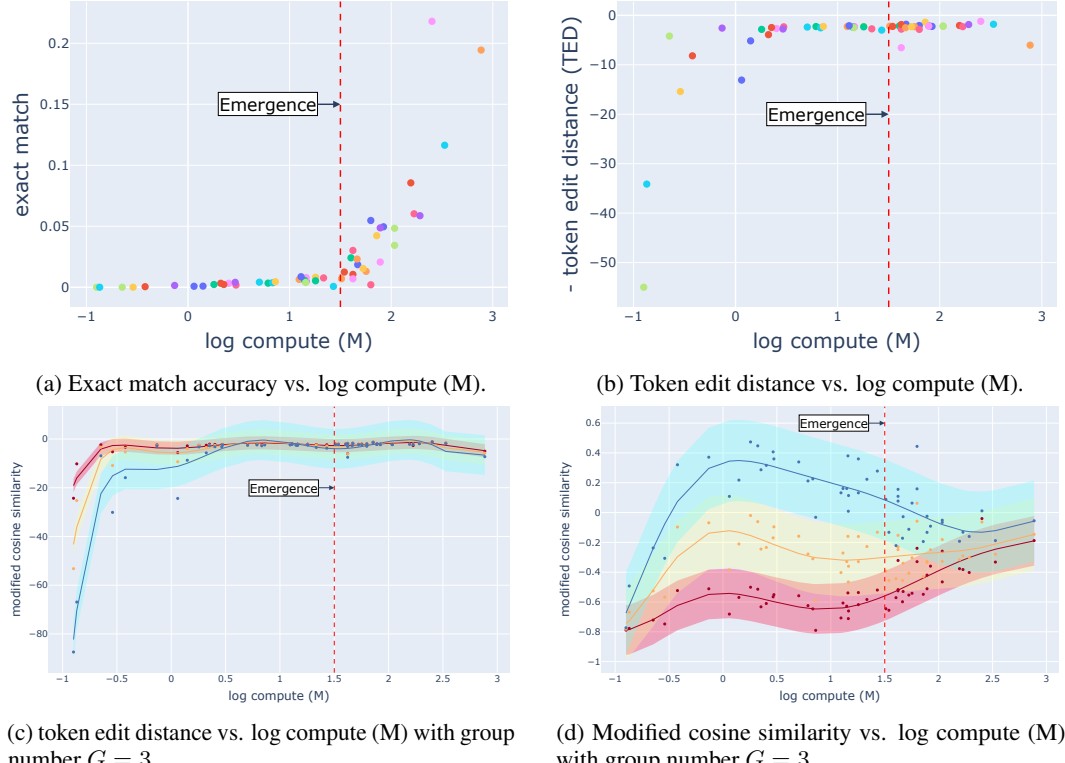

(a) Exact match accuracy vs. log compute (M).

(b) Token edit distance vs. log compute (M).

(c) token edit distance vs. log compute (M) with group number $G = 3$.

(d) Modified cosine similarity vs. log compute (M) with group number $G = 3$.

Figure A26: The exact match accuracy, token edit distance (TED), and cosine similarity score vs. log compute (M) on the word unscramble dataset in BIG-bench (Srivastava et al., 2023).

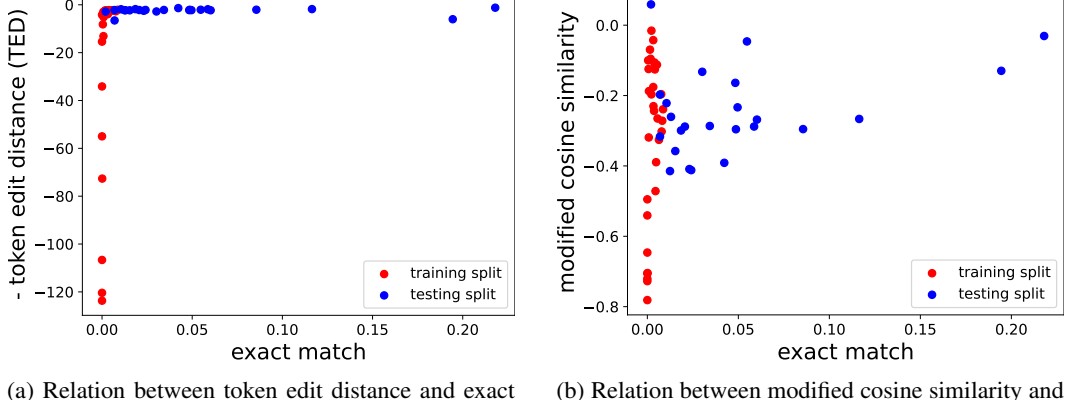

(a) Relation between token edit distance and exact match.

(b) Relation between modified cosine similarity and exact match.

Figure A27: Relations between token edit distance/modified cosine similarity and exact match on the word unscramble dataset in BIG-bench (Srivastava et al., 2023).

# H PRELIMINARY ANALYSIS FOR STRING-MATCH TASKS

This section provides the preliminary analysis for the exact string match tasks.

Fig. A26a shows that the word unscramble dataset in BIG-bench (Srivastava et al., 2023) exhibits emergent abilities under the traditional metric: exact match accuracy. On the other hand, Fig. A26b shows that model performance measured by token edit distance (TED) as discussed in Schaeffer et al. (2024a) improves with scale steadily at first and then exhibit flat scaling.

We argue that TED is not a good measure of progress on a string match task. (1) It does not differentiate between easy and hard questions well. Fig. A26c shows that performance measured by TED on all three question groups is close for all model log computes above $0.5$. A harder group's TED may be higher or lower than an easier group's. (2) It is not very correlated with exact match, the traditional metric that people are probably ultimately interested in ( A27a).

One idea is to measure performance by modified cosine similarity (MCS):

$$MCS = \frac{F(s1) \cdot F(s2)}{\|F(s1)\|\|F(s2)\|} \cdot \mathbb{I}(s1 \subseteq s2),\tag{7}$$

where $s1$ is the model's output string, $s2$ is the answer string, $F(x)$ is CLIP (Radford et al., 2021)'s text encoder to project the string to the vector space, and $\mathbb{I}(x)$ is an indicator function having 1 if every single character of $s1$ is contained in $s2$, otherwise 0. MCS takes values in the interval $[-1, 1]$ and is good at differentiating questions by difficulty levels. Fig. A26d shows that MCS scaling curves of the easy, medium, and hard question groups are clearly ordered.

Interestingly, performance measured by MCS for all question groups exhibits inverted-U scaling followed by steady improvement. The only differences are the model log compute at which scaling reverts from inverse scaling to standard scaling and also how fast performance goes up/down. However, Fig. A27b shows that MCS is also poorly correlated with the exact match. Even if we can precisely predict the MCS of models above the emergence threshold, conversion back to exact match accuracy will be too noisy to be useful. We hope this section illustrates potential avenues for future work.

# I  BROADER IMPACT

## I.1  POTENTIAL POSITIVE IMPACTS

This work identifies U-shaped and inverted-U Scaling of LLM performance once we group questions by difficulty level. We believe this observation can provide the AI community with a deeper understanding of emergent abilities. We also present a forecasting pipeline utilizing the above observation to detect the forthcoming performance soar, the ability of which we believe is crucial in preventing deployment in offensive applications.

## I.2  POTENTIAL NEGATIVE IMPACTS

Given the limitations discussed in Sec. 6, we do not suggest predicting the forthcoming emergent abilities based on merely one of the methods we discuss. Multiple techniques should be used in parallel to prevent possible false positives or false negatives.

