# OpenReview forum: "U-shaped and Inverted-U Scaling behind Emergent Abilities of Large Language Models"
_ICLR.cc/2025/Conference — ICLR 2025 Poster_

### Official Review · Reviewer_K8WA · 2024-10-19

**Soundness:** 3
**Presentation:** 2
**Contribution:** 3
**Rating:** 6
**Confidence:** 3

**Summary:**

By dividing questions according to difficulty levels, the authors observe U-shaped scaling for hard questions and inverted-U scaling for easy questions, followed by steady improvement. Then the authors propose a simple pipeline, called Slice-and-Sandwich, to predict the model performance.

**Strengths:**

1. The inverted-U vs. U-shape scaling seems interesting.

2. The pipeline seems to be novel. Although the basic principle of the method is fairly easy to understand, it substantially outperforms the baseline methods.

3. The experiments are diverse. The authors uses 56 LLMs to evaluate the difficulty levels, and the effectiveness of Slice-and-Sandwich is demonstrated on 6 benchmarks.

**Weaknesses:**

1. The fitted scaling trend is either underestimated or overestimated the actual scaling trend. Although the authors have given explanations, it should be clearly pointed out that why this  underestimation /  overestimation is acceptable. For example, the prediction error of the score of hard problems on Arithmetic benchmark (figure 7) seems actually comparable to the prediction error of the score of all problems on Arithmetic benchmark  (figure 9).

2. The authors provide possible explanations for the inverted-U vs. U-shape scaling. They provide examples of easy/hard problems and give very brief descriptions of the performance of different models. The concepts would be easier to understand if the authors can provide some specific outputs of weak/strong LLMs on easy/hard problems in the appendix.

3. There are some typos in the paper. For example, in all figures, the authors use exactly the same color for the easiest and most difficult problems, making it a bit confusing. In line 484, $F_s^c$ should be $F_e^c$. Also, the layout of the appendix seems weird. For instance, P21 only contains a single huge image.

4. Perhaps the authors should use one sentence in the introduction section (line 53) to explain what is `binary brier score` to make it easier for readers to understand.

**Questions:**

1. I am interested in understanding why averaging the scaling trends of both the easy group and the hard group leads to an accurate prediction of the overall aggregate scaling trend. It's possible that with one prediction being overestimated and the other underestimated, the outcome appears to be coincidentally accurate. The author should provide further (possibly qualitative) explanation to demonstrate that the good result is not a matter of chance. If this question is resolved, I believe the soundness and contribution of the paper can be substantially improved.

2. The authors provide a separate figure for each benchmark. Would it be possible to also provide a figure that includes the results from all benchmarks?

---

> ### Author Response · Authors · 2024-11-19
> **Official Response to Reviewer K8WA**
>
> Thank you for the insightful feedback and comments! We would like to discuss your concerns and how we have addressed them. We have uploaded the revised manuscript.
>
> **1. Why is this underestimation/overestimation acceptable? Why averaging the scaling trends of both the easy group and the hard group leads to an accurate prediction of the overall aggregate scaling trend?**
>     We have revised Sec.5.1 to incorporate this discussion. Below is a detailed explanation:
>     An overestimation of easy questions and an underestimation of hard questions that constitutes a final accurate prediction is not a surprise. We intentionally design and expect Slice-and-Sandwich to produce easy/hard scaling trends that encapsulate the final scaling trends, resulting in decent prediction after averaging (sandwiching) even if underestimation in easy and underestimation in hard question groups happen.
> Specifically, the underestimation for the hard group is acceptable if the easy group is over- or precisely estimated, and vice versa, since averaging (sandwiching) the function will decrease the deviation. Therefore, we want to ensure that the two scaling trends encapsulate the original scaling trend. The polynomial degree selection can achieve this. The degree selection incorporates our prior belief of the fitting power of polynomial regression under different degrees to encapsulate the easy and hard question groups. For instance, if degree=2 is used for hard question groups, one might expect hard question groups to be precise or underestimated since degree=2 has a relatively low power. Then, users can adopt a higher degree for easy question groups, such as 5, to ensure this group is either precise or at least overestimated. A parameter combination that prevents simultaneous under- or over-estimation on both groups should be fine.
> The degree combination of (5, 2) for (easy, hard) question groups may fail in some extreme cases; for instance, both underestimate the scaling trend. However, this case means the emergence is very steep and more drastic than arithmetic and MMLU, where the two have demonstrated quite abrupt emergence.  We further added additional experiments at https://anonymous.4open.science/r/iclr_rebuttal-583C/ regarding (1) Slice-and-Sandwich on non-emergent tasks as in 7. of Official Response to Reviewer nJbX and (2) Slice-and-Sandwich under different group numbers as in 7. of Official Response to Reviewer 9qWW to demonstrate the robustness of this simple yet effective pipeline. We have incorporated (2) to App.F.1.3.
>
> **2. Specific outputs of weak/strong LLMs on easy/hard problems in the appendix**
>     We promise that we will put some examples into the appendix ASAP.
>
> **3. The same color for the easiest and most difficult problems**
>     Please refer to 1.d of Official Responses to Shared Concerns.
>
> **4. The layout of the appendix (Fig. A16 is on a single page)**
>     We have made the appendix more compact by decreasing the size of Fig. A16 so that Fig. A15 and Fig. A16 are now on the same page. Also, Fig. A24 and Fig. A25 are on the same page.
>
> **5. Typo in Eq. (6)**
>     We have revised it. Please see line 422.
>
> **6. Adding one sentence in the introduction section (line 53) to explain what is binary brier score**
>     We have added it. Please see lines 93-96.
>
> **7. Providing a figure that includes the results from all benchmarks**
>     Please refer to 1.e of Official Responses to Shared Concerns.
>
> We also recommend referring to Official Responses to Shared Concerns to see how we have improved the paper presentation, addressed some concerns regarding generalizability (maybe related to soundness), and clarified our contributions.
>
> We hope we have addressed all of your concerns. If any concerns remain, please do not hesitate to let us know. We will respond ASAP.

---

> > ### Comment · Reviewer_K8WA · 2024-11-23
> >
> > Thank you for your response. Although the deeper motivation behind this approach is still not entirely clear, I acknowledge that it appears reasonably effective in the experiments. Additionally, the paper has become much clearer after the revisions. I am happy to update my score.

---

> ### Author Response · Authors · 2024-11-28
> **Official Response to Reviewer K8WA**
>
> As promised, we have uploaded the revised manuscript with the App. E added to include specific outputs of Pythia on some easy/hard problems in MMLU (in line 1457, there is a typo: "...worse than Pythia-160m"->"...worse than Pythia-70m"). We chose Pythia since this model family's log computes are around where the U-shaped and inverted-U scaling of MMLU showcase.
>
> Thank you again for your constructive feedback. We enjoy the engaged discussion.
> We are still open to discussions, such as deeper motivation behind question grouping or Slice-and-Sandwich, if you think it will affect your rating and assessment.

---

### Official Review · Reviewer_9qWW · 2024-10-27

**Soundness:** 2
**Presentation:** 1
**Contribution:** 2
**Rating:** 6
**Confidence:** 3

**Summary:**

This paper investigates the emergent behavior of large language models as their effective size increases. The authors observe that for certain downstream tasks, the continuous metric exhibits a U-shape for easier questions and an inverted U-shape for more difficult questions below the emergence threshold, with steady improvements beyond this threshold. Additionally, they propose a method for forecasting model performance beyond the emergence threshold.

**Strengths:**

1. The paper connects the scaling trend on easy questions with the double descent phenomenon in deep learning is quite interesting.
2. A group-wise scaling law based on difficulty level can offer more detailed insights compared to a single overall scaling curve.

**Weaknesses:**

1. This paper primarily focuses on the continuous Brier score as a metric, which may not fully capture the validity of the observed scaling trends. It is unclear whether these trends are specific to this single metric. To establish this phenomenon as more general, validation across other metrics is needed.
2. The writing and organization of this paper are difficult to follow. For instance, the scaling plots for various LLMs are presented without clearly specifying the types or model families used, as these details are only provided in the appendix. Additionally, the figure depicting the scaling curves lacks clarity; for example, the label '0_1502_brier' in Figure 3 is ambiguous. It appears to represent the index of data points, but using the difficulty level as a label would be more informative.
3. The proposed task in Section 4 is using data collected before the emergence threshold to forecast the occurrence of emergent abilities and the scaling trend beyond this threshold. However, the paper lacks details on how the threshold is selected and how this choice is validated. Furthermore, more explanation is needed to highlight the significance of this task. Why is predicting the scaling curve beyond the emergence threshold based on early performance trends important?
4. In the experiments, a few multiple-choice datasets are utilized to demonstrate the presence of emergent behavior, but they also note that such behavior is absent in other BIG-Bench datasets, as detailed in the appendix. This raises questions about whether the observed emergence is specific to certain datasets rather than a general phenomenon. A more comprehensive analysis is needed to uncover broader patterns and clarify the characteristics of these emergent behaviors—for instance, identifying which types of tasks exhibit this emergence and which do not. This would help determine the conditions under which such emergence is applicable.

**Questions:**

1. The authors appear to categorize difficulty levels as easy, medium, and hard. How might the granularity of these difficulty levels affect the results?
2. In Section 4, the proposed method Slice-and-Sandwich first fits a function to the binary Brier score and then projects the forecasted scaling trend back to accuracy. I am trying to understand the benefit of this additional step. My interpretation is that fitting the binary Brier score curve is more precise, particularly for extrapolation, compared to directly fitting the accuracy curve. Could the authors elaborate on why this approach is preferred over fitting the accuracy curve directly?

---

> ### Author Response · Authors · 2024-11-19
> **Official Response to Reviewer 9qWW**
>
> Thank you for your constructive suggestions! We would like to address your concerns as follows, and we have uploaded the revised manuscript.
>
> **1. Validation of this phenomenon across other metrics**
>     We kindly remind you that App.E has included the accuracy-based observation of this phenomenon. The U-shaped scaling of the accuracy-based procedure is mitigated, while this mitigation is explained well in App.E. The Brier Score and accuracy are the two most iconic metrics for multiple-choice question tasks.
>
> **2. Confusing names for difficulty levels (e.g., 0_1502_brier)**
>     Please refer to 1.c of Official Responses to Shared Concerns.
>
> **3. Scaling plots for LLMs are presented without clearly specifying the types or model families**
>     We currently leave the specification of model families in the appendix due to the page limit. At the same time, we added additional descriptions to Fig. 1 and the text body where we first mentioned Fig. 1 to enhance the clarity and writing flow. Please refer to lines 87-89 and 93.
>
> **4. How the threshold is selected, how is this choice validated?**
>     There is no rigorous mathematical definition of the emergence threshold. Hence, we use 0.1 as the basic unit and eyeball to manually select a proper value that we think splits the emerging/non-emerging region well. Besides the fitting results in the main paper, App.F.1 has also included robustness analysis on different cutoff points for Slice-and-Sandwich. Plus, recognizing a slightly different emergence threshold will not affect our findings of U-shaped & inverted-U scaling. We revised lines 134-136 to make it clearer.
>
> **5. Why is predicting the scaling curve beyond the emergence threshold based on early performance trends important?**
>     Estimating or forecasting the emergence is crucial for AI safety. For instance, when training LLMs, we may want to prevent the unintended realization of LLMs with harmful abilities, such as writing certain malicious code [R1]. Assuming small LLMs on hand cannot achieve such adverse, advanced skills, our scaling law guides us to monitor the abilities of LLMs and proactively design safeguards. We have incorporated the above discussion to lines 45-47.
>
> **6. Whether the observed emergence is specific to certain datasets rather than a general phenomenon**
>     As mentioned in our Introduction, [R2] has shown that many tasks with emergence are multiple-choice question (MCQ) tasks; other some are string-matching tasks. Also, [R3], which is the first work reporting emergent abilities of LLMs, have also documented which tasks have or have no emergence (Though they only used a small number of LLMs, please refer to 2. of Official Responses to Shared Concerns for details). As a result, we selected 6 representative MCQ datasets previously reported to have emergence in the main paper and mentioned string-matching tasks as future direction.
> If you are, in fact, referring to the generalizability of U-shaped vs. inverted-U scaling, please refer to 2. of Official Responses to Shared Concerns.
>
> **7. How might the granularity of these difficulty levels affect the results of slice-and-sandwich?**
>     The granularity of difficulty levels has minor effects on Slice-and-Sandwich since the fitting lines within the same scaling type (e.g., inverted-U) demonstrate a gradual transition, resulting in a similar final scaling law after taking the average. Experimental results are in the folder "slice-and-sandwich/robustness_on_group_number" at https://anonymous.4open.science/r/iclr_rebuttal-583C/ (polynomial degree= 5 & 2 for inverted-U & U-shaped scaling as in the main paper). We have added this robustness analysis on group number to App.F.1.3.
>
> **8. Why don’t we directly fit the accuracy curve?**
>     You are right. As Fig. A21 shows, question groups’ scaling trends on accuracy are mitigated and flatter compared with Brier ones, especially the inverted-U scaling, making function fitting difficult. This is why we fit on the TC Brier Score. Direct accuracy fitting on Persian-QA works since it still shows very strong U-shaped vs. inverted-U when measured by accuracy. Experimental results are in the folder "slice-and-sandwich/fit_on_acc" at https://anonymous.4open.science/r/iclr_rebuttal-583C/ (degree= 5 & 2 for inverted-U & U-shaped scaling as in the main paper).
>
> We recommend referring to Official Responses to Shared Concerns to see how we have improved the paper presentation, addressed some concerns regarding generalizability (maybe related to soundness), and clarified our contributions.
>
> Hope we have addressed all of your concerns! If you have any remaining, please do not hesitate to let us know. We will respond ASAP.
>
> [R1] From Text to MITRE Techniques: Exploring the Malicious Use of Large Language Models for Generating Cyber Attack Payloads (Arxiv 2023)
> [R2] Are emergent abilities of large language models a mirage? (NeurIPS’24)
> [R3] Emergent Abilities of Large Language Models (TMLR)

---

> > ### Comment · Reviewer_9qWW · 2024-11-24
> >
> > Thank you for the response. It effectively addresses my concern, and I will adjust the score accordingly

---

> > > ### Author Response · Authors · 2024-12-01
> > > **Official Response to Reviewer 9qWW**
> > >
> > > Thank you again for the suggestions that enable us to improve this work!

---

### Official Review · Reviewer_nJbX · 2024-11-02

**Soundness:** 2
**Presentation:** 1
**Contribution:** 2
**Rating:** 6
**Confidence:** 3

**Summary:**

The paper investigates explanations for LLM emergent abilities (sharp transitions from chance-level to high performance as a function of model size) and proposes a method to predict emergent performance. They show that on certain task, when questions are group by difficulty, two distinct scaling trends appear: U-shaped scaling for hard questions and inverted U-shaped followed by improvements for easy questions. This motivated a method for capabilities forecasting: fitting polynomial regression separately for easy and hard questions and then aggregating the results.

**Strengths:**

1. The paper addresses a very timely topic: the science of LLM evals and predictive evals. It is relevant for scientific understanding of frontier LLMs as well as for AI policy (informing capability thresholds and buffers in responsive scaling policies).
2. The paper described an interesting insight that easy and hard questions have different scaling trends and that this could explain sharp transitions of aggregate performance

**Weaknesses:**

1. I found the paper hard to follow. I think the clarity of writing and figures could be substantially improved. Some concrete suggestions:
(a) the name "binary Brier score" is very confusing. What's binary about it? It's continuous for each question.
(b) I think it's very confusing to say Brier score is "noisy due to insufficient confidence calibrations" (line 161). What does it mean to be sufficient? Why is this noise? Why "calibrations" in plural?
(c) what are "available choices' (line 159)? Is it the same as classes (line 139)?
(d) names for difficulty levels (e.g. 0_1502_brier) are confusing. They're okay in Python code, but I'd expect something more readable and abstracting away from distracting details (e.g. I don't really care it's 1502 questions).
(e) The colors on plot could match the semantics of difficulties, e.g. you could use a continuous color map (e.g. red-blue transition to represent easy-hard transition).
(f) plots for different datasets could be next to each other
(g) "Slide-and-sandwich approach allows the data to speak for itself" is misleading. The whole point is that you bake in more assumptions in the method.
2. The name "effective compute" is somewhat misleading if it's just log compute. Ideally, effective compute could be something like g-factor [1] or the first principal component from observational calling laws [2, 3]. It would be great to use that as a metric. Or at the very least, it should be just called "log compute".
3. Authors only evaluate their methods on 3 datasets (plus 3 more in the appendix). I don't think that's enough to justify the claim that they found a general method for capabilities forecasting and a general explanations of emergent features. BIG-Bench includes more than 200 tasks.
4. Relatedly, all of them do show emergent capabilities. I wonder how Slice-and-sandwich methods behaves on tasks that do not show emergent capabilities: does it falsely predict them? I think this is an important point: so far the authors only presented a method for emergent capabilities forecasting conditional on the presence of emergent capabilities. But in practice we don't know that in advance.
5. Overall, I'm somewhat underimpressed by predictive performance and I worry that the authors overfit to a particular set of 6 tasks. If find some patterns in a dataset and bake in this pattern as an inductive biases of your method (polynomial regression), surely you'll improve predictive performance. But will those inductive biases generalize to entirely new tasks you never looked at? The authors provide no evidence for that (unless I'm missing something, happy to be proved wrong).
6. I'm not super convinced this is a general explanation of emergent capabilities. Maybe others have different scaling trends? The whole point that a superposition of U-shaped and inverted-U shaped scaling explains emergent capabilities is also somewhat qualitative. It's not starkly clear from the plots, e.g. MMLU does not seem consistent with this explanation.
7. Consider discussing [4] as related work

[1] Unveiling the general intelligence factor in language models: A psychometric approach

[2] Observational Scaling Laws and the Predictability of Language Model Performance

[3] Safetywashing: Do AI Safety Benchmarks Actually Measure Safety Progress?

[4] The Quantization Model of Neural Scaling

**Questions:**

1. How were the 6 tasks you experiment on selected? What were the criteria and at what point were they selected?
2. What's the functional form of the polynomials you fit (line 416)?

---

> ### Author Response · Authors · 2024-11-19
> **Official Response to Reviewer nJbX**
>
> Thank you for your valuable feedback and comments! We would like to address your questions and concerns below. We have uploaded the revised manuscript.
>
> **1. What does "available choices" (in original line 159) mean?**
>     "available choices" is exactly "all classes" of a multiple-choice question. In case "choices" is confusing, we have revised "available choices" to "all classes" in Sec.2.
>
> **2. Confusing sentence: "noisy due to insufficient confidence calibrations" (in original line 161)**
>     The footnote is indeed ambiguous. We deleted and merged it into lines 148-149 and referenced [R2]. We rephrased Lines 152-161 to make them clearer.
>
> **3. Misleading sentence: "Slide-and-sandwich approach allows the data to speak for itself"**
>     We revised lines 518-520 to improve the statement as suggested.
>
> **4. Regarding the remaining points in 1. and 2. of your review**
>     Please refer to 1. of Official Responses to Shared Concerns.
>
> **5. Using Slice-and-Sandwich methods on non-emergent tasks**
>     Slice-and-Sandwich doesn’t need to be used on tasks that have experienced an increasing learning curve. Given some small-scale LLMs, such as models with M=-1, 0, or 1, we should have known whether the performance stagnates. If not, an ordinary Sigmoid-based scaling law is good enough under this condition. If so, the question grouping, U-shaped vs. inverted-U scaling trends, and Slice-and-Sandwich method can be applied to provide insights into the learning process of LLMs behind the stagnation.
>
> However, Slice-and-Sandwich can indeed serve as a normal scaling law for early- and middle-stage performance estimation if we consider the medium question group, as the medium question group of non-emergent tasks, such as hellaswag, demonstrates apparent performance increase with scales. We have put experimental results of Slice-and-Sandwich on hellaswag, arc, and abstract_narrative_understanding to the folder "slice-and-sandwich/performance on non-emergent tasks" at https://anonymous.4open.science/r/iclr_rebuttal-583C/, with OLS used for fitting the scaling trend of medium question group and the Brier-based scaling is acquired by taking the average of easy, medium, and hard trends. Slice-and-sandwich is generally better than a Sigmoid-based baseline. Notably, fitting and taking the average of easy, medium, and hard trends will only have minor effects on the results of emergent tasks, such as MMLU, arithmetic, and Persian-QA, since the stagnant performance on medium question groups leads to flat, nearly constant OLS.
>
> **6. How were the 6 tasks selected?**
>     Please refer to 2.a of Official Responses to Shared Concerns.
>
> **7. Generalizability of Slice-and-Sandwich and U-shaped vs. inverted-U scaling. (Maybe others have different scaling trends?)**
>     We value this concern a lot. In short, the U-shaped or inverted-U are prevalent. They may even exist in non-emergent tasks, such as causal judgment in Big-Bench. While they may not coexist in a dataset, for instance, crass ai in Big-Bench has an overall U-shaped performance, and its easy, medium, and hard question groups are all U-shaped. The six datasets in our papers are representative multiple-choice question datasets with apparent emergence, and they all, more or less, have the two kinds of scaling trends. Detailed explanations are in 2. of Official Responses to Shared Concerns.
>
> **8. What is the functional form of the polynomials you fit (in original line 416)?**
>     We added the polynomial degree used in the main paper at line 470. The functional form of polynomial regression is $y = \beta_0 + \beta_1 x + \beta_2 x^2 + \beta_3 x^3 + \cdots + \beta_n x^n + \epsilon$, where $n$ is the degree.
>
> **9. Discussing [R1] as related work**
>     We have added it. Please see lines 42-43.
>
> We hope we have addressed all of your concerns. If you have any remaining concerns, please do not hesitate to let us know. We will respond ASAP.
>
> [R1] The Quantization Model of Neural Scaling (NeurIPS 2023)
> [R2] Can Multiple-choice Questions Really Be Useful in Detecting the Abilities of LLMs? (LREC-COLING 2024)

---

> > ### Comment · Reviewer_nJbX · 2024-11-24
> >
> > Thanks for the detailed response! I updated my score.

---

> > > ### Author Response · Authors · 2024-12-01
> > > **Official Response to Reviewer nJbX**
> > >
> > > Thank you again for your constructive feedbacks that lead to the large improvement of our revised manuscript!

---

### Official Review · Reviewer_bboC · 2024-11-03

**Soundness:** 3
**Presentation:** 4
**Contribution:** 4
**Rating:** 8
**Confidence:** 4

**Summary:**

The paper connects two well-known phenomena in deep learning, namely emergent capabilities and inverse scaling. By breaking down the scaling curve to different difficulty levels, the authors show that LLM skill emergence can be the result of an inverted-U-shaped scaling on easy questions, where the easy-questions curve cancels out the scaling on hard questions. The authors propose a method to predict emergent scaling by fitting sub-groups of the dataset separately.

**Strengths:**

- The main result of this paper connects two important topics in the field, inverse scaling and emergence, making it a very relevant research topic, especially since neither phenomenon is well understood.
The results on MMLU and Persian-QA are clear enough, in my opinion, to establish the existence of a connection between U-shaped scaling and emergence. Although Big-bench arithmetic and the appendix results are less clear, they do not diminish from the other results. Even if the easy-hard data split failed to visualize inverse scaling in the latter experiments, a different split might still show similar curves to MMLU.

- The proposed fitting method, slice-and-sandwich, showcases how the documented phenomenon can be used to predict emergence in practice.

- The paper is clearly written and easy to follow.

**Weaknesses:**

- While the fitting method is an overall positive addition to the paper, it is less convincing than the scaling results (which are, for me, the main result of the paper). The fits shown in the main section use knowledge of the point where emergence starts, making them of little value, since predicting that point is the main reason why we care about fitting emergence.
Appendix F2 fixes that issue by showing fits on random sections of the scaling curve before emergence, but these fits are not so convincing as a reliable method to predict the point of emergence.

- Appendix E shows the scaling curves when plotting accuracy, and attributes the difference from the main results to accuracy being a 'crude measure', which is not a satisfactory explanation to me.

**Questions:**

- Minor correction: In the definition of $M$ (line 128), $N$ and $D$ are mixed up.

- Naming $M$ 'effective model size' is confusing for the reader, since model size is $N$ while $M$ is actually the log of compute $C$. Maybe a better name would be 'effective compute' or just 'log compute?

---

> ### Author Response · Authors · 2024-11-19
> **Official Response to Reviewer bboC**
>
> Thank you for appreciating our work! Below, we discuss your concerns and how we have addressed them. We have uploaded the revised manuscript.
>
> **1. Reliability of Slice-and-Sandwich**
>     Slice-and-Sandwich directly results from our observations of different scaling trends on different question groups. Its most significant value is to give the first positive vote of "Yes" on the ongoing discussion of predictability of emergent ability in the AI community. We admit that Slice-and-Sandwich has room to improve, probably using more sophisticated fitting methods. We look forward to future efforts by the community to develop more advanced techniques to forecast the abrupt ability transition of LLMs better.
>
> **2. Explaining the difference between main results and accuracy-based results**
>     We rephrased some descriptions and added more detailed explanations in lines 1312-1320.
>
> **3. N and D typo correction**
>     Corrected. Please see lines 134-135.
>
> **4. Ambiguous name of effective model size**
>     Please refer to 1.a of Official Response to Shared Concerns.
>
>
> We hope we have addressed all of your concerns. If you have any remaining concerns, please do not hesitate to let us know. We will respond ASAP.

---

> > ### Comment · Reviewer_bboC · 2024-11-24
> >
> > Thank you for your response. The revised version is clearer, especially the plots which are now more readable. I might suggest replacing the 'difficulty level' sidebar with a colorbar, since the repeating text there is not necessary other than stating that the lines are fits.
> >
> > I will maintain my score of 8. I want to reiterate that this paper's merit, in my opinion, stems mostly from the empirical results, and not from Slice-and-Sandwich. The idea behind Slice-and-Sandwich is good, but its results are underwhelming, especially those presented in the main text.

---

> > > ### Author Response · Authors · 2024-11-28
> > > **Official Response to Reviewer bboC**
> > >
> > > We have uploaded the revised manuscript, replacing the figures' 'difficulty level' sidebar with a colorbar. Thank you again for your constructive feedback. We enjoy the engaged discussion.

---

### Author Response · Authors · 2024-11-19
**Official Response to Shared Concerns (1)**

This response is for shared concerns about paper writing and experiments.

1. Paper writing.

    **a. Ambiguous name of effective model size**
        As suggested, we revised the term to "log compute" in all figures, tables, text body, and appendix.

    **b. Ambiguous name of binary Brier Score**
       We revised the term to "Target-Conditioned (TC) Brier Score" to make it clear in all figures, tables, text body, and appendix. TC means our Brier Score takes the **target** label **conditioned** on the probability sum of all classes.

    **c. Ambiguous names for difficulty levels (e.g. 0_1502_brier)**
       We revised their names to difficulty levels 1, 2, ..., and 10 and have marked 1 as easiest and 10 as hardest in all figures, tables, text body, and appendix.

    **d. Regarding the color map for different question groups**
       We revised the color of different difficulties to gradual transitions from red to blue, as suggested by Reviewer nJbX, including all figures in the main paper and appendix.

    **e. A plot that different datasets are next to each other**
       We added it as Fig. 4 in the revised manuscript and added proper descriptions for it in Sec.2.3.

    **f. Self-correction**
       Fig.1(b) in the original submission was the un-redistributed Brier Score version that didn’t apply Eq. (4). We have fixed it.

2. Detailed experiment settings
    **a. How are these six datasets selected?**
        The 6 datasets are chosen due to their representativeness as MCQ tasks with ability emergence. We aim to choose MCQ datasets with apparent emergence, i.e., model performance is flat across scales and soars at a certain threshold.
        The MMLU was first chosen since it is a classic LLM multiple-choice question (MCQ) dataset. Arithmetic and Persian-QA (MCQ version) are subsequently selected from Fig.2 of [R1] (Word unscramble also has an apparent ability emergence, while it didn't have MCQ implementation). This is how we select the 3 datasets.
        We further tested more tasks (datasets) in Big-Bench. [R1] has documented 66 Big-Bench datasets with emergence. However, they only used PaLM, GPT-3, or LaMDA to test the scaling trend. When evaluating with 56 LLMs, we found some tasks were, in fact, non-emergent. They have different scaling trends, such as steady improvement or even U-shaped scaling. Plus, some tasks in Big-Bench have a tiny number of questions, resulting in reduced statistical reliability, especially when they are grouped into subsets.
        Specifically, we have examined an additional 17 tasks in Big-Bench with 56 LLMs, totaling 20 tasks. 5 of them only have string-matching task implementation. Evaluation order is simply alternating between [R1]’s App.E.2 and App.E.3. The 3 datasets in our App. are the **largest** datasets that are (1) MCQ tasks and (2) still have clear emergence when evaluated with 56 LLMs. Experimental results of the remaining 9 datasets, which are either MCQ or True/False questions, are in https://anonymous.4open.science/r/iclr_rebuttal-583C/. Detailed information of the 17 tasks following our evaluation order is as follows:

### Table 1: Examination of 17 Big-Bench tasks. Tasks marked with strikethrough only have string-matching task implementation; tasks marked with bold are those adopted in our paper. Our experiments stopped at Hindu knowledge.
| Task name | Clear emergence | Question number > 100 | MCQ | Level of U-shaped vs. inverted-U Scaling | Sec. in [R1] |
|-------------|-------------|--------------|---------|--------------|--------|
| analytic entailment       | inverted-U    | &#10008;      | &#10008; | Medium | App.E.2 |
| anachronisms       | &#x2714;   | &#x2714;      | &#10008; | Medium | App.E.3 |
| **analogical similarity**      | &#x2714;   | &#x2714;      | &#x2714; | Medium | App.E.3 |
| ~~ascii word selection~~      | N/A  | N/A      | &#10008; | N/A | App.E.3 |
| ~~auto debugging~~      | N/A  | N/A      | &#10008; | N/A | App.E.3 |
| ~~codenames~~      | N/A  | N/A      | &#10008; | N/A | App.E.2 |
| common morpheme     | &#10008;  | &#10008;      | &#x2714; | Low-to-no | App.E.2 |
| causal judgment     | &#10008;  | &#x2714;     | &#10008; | Strong | App.E.3 |
| code line description     | &#x2714;  | &#10008;     | &#x2714; | Medium | App.E.3 |
| **conceptual combinations**     | &#x2714;  | &#x2714;     | &#x2714; | Strong | App.E.3 |
| crass ai     | U-shaped  | &#10008;     | &#x2714; | All U-shaped | App.E.3 |
| ~~cryptonite~~     | N/A  | N/A     | &#10008; | N/A | App.E.3 |
| cs algorithms     | &#x2714;  | &#x2714;     | &#10008; | No | App.E.3 |
| fact checker     | &#x2714;  | &#x2714;     | &#10008; | Inverted | App.E.2 |
| figure of speech detection     | &#10008;  | &#10008;     | &#x2714; | No | App.E.2 |
| ~~gender inclusive sentences german~~     | N/A  | N/A     | &#10008; | N/A | App.E.2 |
| **Hindu knowledge**     | &#x2714;  | &#x2714;     | &#x2714; | Strong | App.E.2 |

---

> ### Author Response · Authors · 2024-11-19
> **Official Response to Shared Concerns (2)**
>
> 2. Detailed experiment settings (Continued)
>     **a. How are these six datasets selected? (Continued)**
>     Note that even non-emergent tasks can demonstrate U-shaped vs. inverted-U scaling, such as causal judgment, while the two trends do not offset each other because they are at different levels. In addition, some tasks, such as crass ai, have a unique overall scaling trend, such as U-shaped scaling, where easy, medium, and hard question scaling trends all demonstrate U-shape.
>     We believe a thorough, large-scale evaluation of all 66 datasets with question difficulty grouping will be a promising direction for future work and will bring many remarkable insights into previously reported interesting LLM learning dynamics. While this is beyond this work’s scope, where we put our emphasis on (1) introducing question grouping by difficulty level, (2) using this technique to explore U-shaped vs. inverted-U scaling to explain ability emergence, and (3) using this observation to realize forecasting of ability emergence.
>
>     **b. Generalizability of our proposed phenomenon**
>     Through **a.**, we believe the 6 representative MCQ datasets featuring clear emergence and large enough sizes have been convincing enough. Indeed, it might be hard to claim every MCQ dataset with emergence must have substantial inverted-U vs. U-shaped scaling trends (we added this to the Conclusions and Limitations in the revised manuscript)—that’s too ideal—since the level and clarity of emergence, U-shaped, and inverted-U scaling do vary in each dataset under influence by many potential factors like question types, dataset sizes, and even prompt formats; while we believe the current experiments and the paper overall are adequate to document U-shaped vs. inverted-U as a possible cause behind many MCQ datasets that still have emergence phenomenon under our large-scale, thorough evaluation.
>
>     **c. We want to clarify our contributions here**
>     As pointed out by Reviewer bboC:
>     Main contribution: we identify a unique scaling pattern, the U-shaped vs. inverted-U scaling, behind various MCQ tasks with emergent abilities, providing a detailed possible explanation for this notorious phenomenon.
> 	Auxiliary contribution: the identification of U-shaped vs. inverted-U scaling intuitively leads to Slice-and-Sandwich, the first (positive) vote for the long debate of predictability of emergent abilities. The prediction of ability emergence is critical to understanding the learning dynamics of LLMs, monitoring certain abilities of LLMs, and proactively designing safeguards for potentially harmful applications.
>
> [R1] Emergent Abilities of Large Language Models (TMLR)
> [R2] Are emergent abilities of large language models a mirage? (NeurIPS’24)

---

### Meta-Review · Area_Chair_9qcb · 2024-12-19

**Metareview:**

The paper connects two well-known phenomena in deep learning, namely emergent capabilities and inverse scaling. And it proposes a simple yet effective pipeline, called Slice-and-Sandwich, to predict both the emergence threshold and model performance beyond the threshold. Although there were some disagreements, the reviewers generally found that the paper was well-written, thoroughly experimented, and presented novel ideas. After the rebuttal, the clarity and solidity of the paper was improved, and it received unanimous recognition and higher scores from the reviewers. AC believes these evaluations were well-considered and credible, therefore recommends accepting this paper.

**Additional Comments On Reviewer Discussion:**

All reviewers have responded to the authors' rebuttal and explicitly indicated that the authors have addressed their concerns and are willing to support the acceptance of this paper.

---

### Decision · Program_Chairs · 2025-01-22

Accept (Poster)